# A GENERATIVE MODEL FOR MOLECULAR DISTANCE GEOMETRY

## ABSTRACT

Computing equilibrium states for many-body systems, such as molecules, is a long-standing challenge. In the absence of methods for generating statistically independent samples, great computational effort is invested in simulating these systems using, for example, Markov chain Monte Carlo. We present a probabilistic model that generates such samples for molecules from their graph representations. Our model learns a low-dimensional manifold that preserves the geometry of local atomic neighborhoods through a principled learning representation that is based on Euclidean distance geometry. In a new benchmark for molecular conformation generation, we show experimentally that our generative model achieves state-of-the-art accuracy. Finally, we show how to use our model as a proposal distribution in an importance sampling scheme to compute molecular properties.

## 1 INTRODUCTION

Over the last few years, many highly-effective deep learning methods generating small molecules with desired properties (e.g., novel drugs) have emerged (Gómez-Bombarelli et al., 2018; Segler et al., 2018; Dai et al., 2018; Jin et al., 2018; Bradshaw et al., 2019a; Liu et al., 2018; You et al., 2018; Bradshaw et al., 2019b). These methods operate using graph representations of molecules in which nodes and edges represent atoms and bonds, respectively. A representation that is closer to the physical system is one in which a molecule is described by its geometry or *conformation*. A conformation $\mathbf{x}$ of a molecule is defined by a set of atoms $\{(\epsilon_i, \mathbf{r}_i)\}_{i=1}^{N_v}$, where $N_v$ is the number of atoms in the molecule, $\epsilon_i \in \{\mathrm{H, C, O}, ...\}$ is the chemical element of the atom $i$, and $\mathbf{r}_i \in \mathbb{R}^3$ is its position in Cartesian coordinates. Importantly, the relative positions of the atoms are restricted by the bonds in the molecule and the angles between them. Due to thermal fluctuations resulting in stretching of and rotations around bonds, there exist infinitely many conformations of a molecule. A molecule's graph representation and a set of its conformations are shown in Fig. 1. Under a wide range of conditions, the probability $p(\mathbf{x})$ of a conformation $\mathbf{x}$, is governed by the Boltzmann distribution and is proportional to $\exp\{-E(\mathbf{x})/k_B T\}$, where $E(\mathbf{x}) \in \mathbb{R}$ is the conformation's energy, $k_B$ is the Boltzmann constant, and $T$ is the temperature.

To compute a molecular property for a molecule, one must sample from $p(\mathbf{x})$. The main approach is to start with one conformation and make small changes to it over time, e.g., by using Markov chain Monte Carlo (MCMC) or molecular dynamics (MD). These methods can be used to accurately sample equilibrium states of molecules, but they become computationally expensive for larger ones (Shim & MacKerell, 2011; Ballard et al., 2015; De Vivo et al., 2016). Other heuristic approaches exist in which distances between atoms are set to fixed idealized values (Havel, 2002; Blaney & Dixon, 2007). Several methods based on statistical learning have also recently been developed to tackle the issue of conformation generation. However, they are mainly geared towards studying proteins and their folding dynamics (AlQuraishi, 2019). Some of these models are not targeting a distribution over conformations but the most stable folded configuration (Evans et al., 2018; Ingraham et al., 2019), while others are not transferable between different molecules (Lemke & Peter, 2019; Noé et al., 2019).

This work includes the following key contributions:

- We introduce a novel probabilistic model for learning conformational distributions of molecules with graph neural networks.

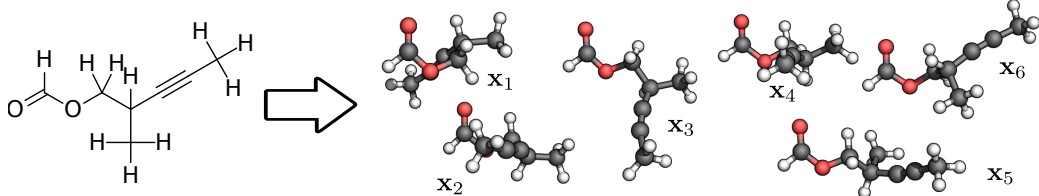

Figure 1: Standard graph representation of a molecule (left) with a set of possible conformations $\{\mathbf{x}_i\}$ (right). Hydrogen (H), carbon (C), and oxygen (O) atoms are colored white, gray, and red, respectively. Conformations feature the same atom types and bonds but the atoms are arranged differently in space. These differences arise from rotations around and stretching of bonds in the molecule.

- We create a new, challenging benchmark for conformation generation, which is made publicly available. To the best of our knowledge, this is the first benchmark of this kind.
- By combining a conditional variational autoencoder (CVAE) with an Euclidean distance geometry (EDG) algorithm we present a state-of-the-art approach for generating one-shot samples of molecular conformations for unseen molecules that is independent of their size and shape.
- We develop a rigorous experimental approach for evaluating and comparing the accuracy of conformation generation methods based on the mean maximum deviation distance metric.
- We show how this generative model can be used as a proposal distribution in an importance sampling (IS) scheme to estimate molecular properties.

## 2 METHOD

Our goal is to build a statistical model that generates molecular conformations in a one-shot fashion from a molecule's graph representation. First, we describe how a molecule's conformation can be represented by a set of pairwise distances between atoms and why this presentation is advantageous over one in Cartesian coordinates (Section 2.1). Second, we present a generative model in Section 2.2 that will generate sets of atomic distances for a given molecular graph. Third, we explain in Section 2.3 how a set of predicted distances can be transformed into a molecular conformation and why this transformation is necessary. Finally, we detail in Section 2.4 how our generative model can be used as a proposal distribution in an IS scheme to estimate molecular properties.

### 2.1 EXTENDED MOLECULAR GRAPHS AND DISTANCE GEOMETRY

In this study, a molecule is represented by an undirected graph which is defined as a tuple $\mathcal{G} = (V, E)$. $V = \{v_i\}_{i=1}^{N_v}$ is the set of nodes representing atoms, where each $v_i \in \mathbb{R}^{F_v}$ holds atomic attributes (e.g., the element type $\epsilon_i$). $E = \{(e_k, r_k, s_k)\}_{k=1}^{N_e}$ is the set of edges, where each $e_k \in \mathbb{R}^{F_e}$ holds an edge's attributes (e.g., the bond type), and $r_k$ and $s_k$ are the nodes an edge is connecting. Here, $E$ represents the molecular bonds (and the auxiliary edges which are explained below) in the molecule.

We assume that, given a molecular graph $\mathcal{G}$, one can represent one of its conformations $\mathbf{x}$ by a set of atomic distances $\mathbf{d} = \{d_k\}_{k=1}^{N_e}$, where $d_k = |\mathbf{r}_{r_k} - \mathbf{r}_{s_k}|$ is the Euclidean distance between the positions of the atoms $r_k$ and $s_k$ in this conformation. As the set of edges between the bonded atoms ($E_{\text{bond}}$) alone would not suffice to describe a conformation, we expand the traditional graph representation of a molecule by adding *auxiliary* edges. Auxiliary edges between atoms that are second neighbors in the original graph fix angles between atoms, and those between third neighbors fix dihedral angles (denoted $E_{\text{angle}}$ and $E_{\text{dihedral}}$, respectively). In this work, $E_{\text{angle}}$ consists of edges between all second neighbors in the original graph. Edges between third neighbors are added according to a heuristic (see Appendix A.1). From now on we are always referring to this extended molecular graph when talking about molecular graphs. In Fig. 2, the process of extending the molecular graph and the extraction of $\mathbf{d}$ from $\mathbf{x}$ and $\mathcal{G}$ are illustrated.

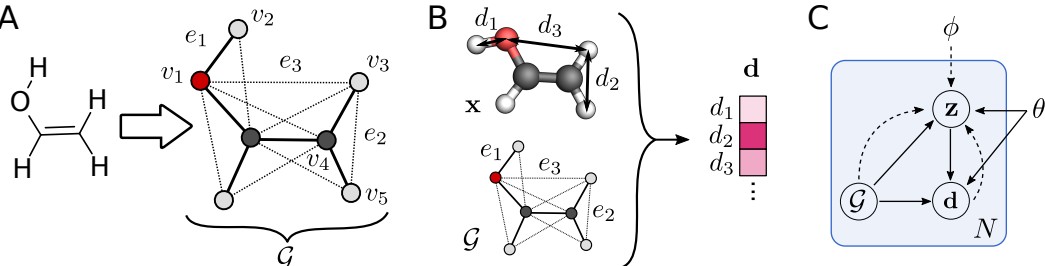

Figure 2: A) The structural formula of a molecule is converted to an extended molecular graph $\mathcal{G}$ consisting of nodes representing atoms (circles, e.g., $v_1$) and edges representing molecular bonds (solid lines, e.g., $e_1 \in E_{\text{bond}}$) and auxiliary edges (dotted lines, e.g., $e_2 \in E_{\text{angle}}$ and $e_3 \in E_{\text{dihedral}}$). B) The distances $\mathbf{d}$ are extracted from a conformation $\mathbf{x}$ based on the edges $E$. C) Graphical model of the variational autoencoder: generative model $p_\theta(\mathbf{d}|\mathbf{z}, \mathcal{G})p_\theta(\mathbf{z}|\mathcal{G})$ (solid lines) and variational approximation $q_\phi(\mathbf{z}|\mathbf{d}, \mathcal{G})$ (dashed lines).

A key advantage of a representation in terms of distances is its invariance to rotation and translation; by contrast, Cartesian coordinates depend on the (arbitrary) choice of origin, for example. In addition, it reflects pair-wise physical interactions and their generally local nature. Auxiliary edges can be placed between higher-order neighbors depending on how far the physical interactions dominating the potential energy of the system reach.

We have a set of $N_{\mathcal{G}}$ molecular graphs $\{\mathcal{G}_l\}_{l=1}^{N_{\mathcal{G}}}$. Further, for each $\mathcal{G}_l$, we have $S_l$ conformational samples $\{\mathbf{x}_{l,j}\}_{j=1}^{S_l}$ from the ground-truth distribution resulting in $S_l$ sets of distances $\{\mathbf{d}_{l,j}\}_{j=1}^{S_l}$. With this data, we will train a generative model which we detail in the following section.

## 2.2 GENERATIVE MODEL

We employ a CVAE (Kingma & Welling, 2014; Pagnoni et al., 2018) to model the distribution over distances $\mathbf{d}$ given a molecular graph $\mathcal{G}$. A CVAE first encodes $\mathcal{G}$ together with $\mathbf{d}$ into a latent space $\mathbf{z} \in \mathbb{R}^{kN_v}$, where $k \in \mathbb{N}^+$, with an encoder $q_\phi(\mathbf{z}|\mathbf{d}, \mathcal{G})$. Subsequently, the decoder $p_\theta(\mathbf{d}|\mathbf{z}, \mathcal{G})$ decodes $\mathbf{z}$ back into a set of distances. A graphical model is shown in Fig. 2 C).

A conformation has, in general, $3N_v - 6$ spatial degrees of freedom (dofs): one dof per spacial dimension per atom minus three translational and three rotational dofs. Therefore, the latent space should be proportional to the number of atoms in the molecule. In addition, the latent space should be smaller than $3N_v$ as it is the role of the encoder to project the conformation into a lower-dimensional space. As a result, we set $k = 1$ to avoid overfitting.[1]

Here, $q_\phi(\mathbf{z}|\mathbf{d}, \mathcal{G})$ and $p_\theta(\mathbf{d}|\mathbf{z}, \mathcal{G})$ are Gaussian distributions, the mean and variance of which are modeled by two artificial neural networks. At the center of this model are message-passing neural networks (MPNNs) (Gilmer et al., 2017) with multi-head attention (Veličković et al., 2018). In short, an MPNN is a convolutional neural network that allows end-to-end learning of prediction pipelines whose inputs are graphs of arbitrary size and shape. In a convolution, neighboring nodes exchange so-called messages between neighbors to update their attributes. Edges update their attributes with the features of the nodes they are connecting. The MPNN is a well-studied technique that achieves state-of-the-art performance in representation learning for molecules (Kipf & Welling, 2017; Duvenaud et al., 2015; Kearnes et al., 2016; Schütt et al., 2017b; Gilmer et al., 2017; Kusner et al., 2017; Bradshaw et al., 2019a).

In the following, we describe the details of the model.[2] In Fig. 3, an illustration of the model is shown. In the encoder $q_\phi(\mathbf{z}|\mathbf{d}, \mathcal{G})$, each $d_k$ is concatenated with the respective edge feature $e_k$ to give $e'_k \in \mathbb{R}^{F_e+1}$. Then, each $v_i$ and each $e'_k$ are passed to $F_{\text{enc},v}$ and $F_{\text{enc},e}$ (two multilayer perceptrons,

---

[1]Experiments showed that our model performs similarly with a latent space of $\mathbb{R}^{2N_v}$ and overfits with latent spaces of $\mathbb{R}^{3N_v}$.

[2]The full model (including all parameters) is available online `https://figshare.com/s/1b42bf865bd78c457354`

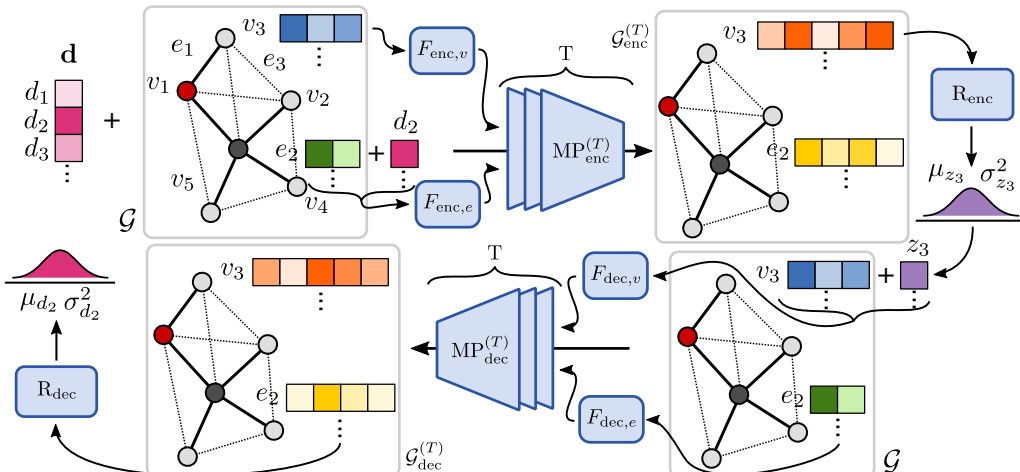

Figure 3: The molecular graph $\mathcal{G}$ together with the distances $\mathbf{d}$ are passed through the model consisting of an encoder $q_\phi(\mathbf{z}|\mathbf{d}, \mathcal{G})$ and a decoder $p_\theta(\mathbf{d}|\mathbf{z}, \mathcal{G})$. See the main text for details.

MLPs), respectively, to give $\mathcal{G}_{\text{enc}}^{(0)}$, where $\mathcal{G}_{\text{enc}}^{(t)} = (\{v_{i,\text{enc}}^{(t)}\}_{i=1}^{N_v}, \{(e_{k,\text{enc}}^{(t)}, r_k, s_k)\}_{k=1}^{N_e}), v_{i,\text{enc}}^{(t)} \in \mathbb{R}^{L_v}$, and $e_{k,\text{enc}}^{(t)} \in \mathbb{R}^{L_e}$. Then, $T$ MPNNs of depth 1, $\{\text{MP}_{\text{enc}}^{(t)}\}_{t=1}^{T}$, are consecutively applied to obtain $\mathcal{G}_{\text{enc}}^{(T)}$. Finally, the read-out function $R_{\text{enc}}$ (an MLP) takes each $v_{i,\text{enc}}^{(T)}$ to predict the mean $\mu_{z_i} \in \mathbb{R}$ and the variance $\sigma_{z_i}^2 \in \mathbb{R}$ of the Gaussian distribution for $z_i$. The so-called reparametrization trick is employed to draw a sample for $z_i$. In summary,

$$v_{i,\text{enc}}^{(0)} = F_{\text{enc},v}(v_i), \quad e_{k,\text{enc}}^{(0)} = F_{\text{enc},e}(e_i'),$$
$$\mathcal{G}_{\text{enc}}^{(1)} = \text{MP}_{\text{enc}}^{(0)}(\mathcal{G}_{\text{enc}}^{(0)}), \quad \mathcal{G}_{\text{enc}}^{(t+1)} = \text{MP}_{\text{enc}}^{(t)}(\mathcal{G}_{\text{enc}}^{(t)}), \quad \mathcal{G}_{\text{enc}}^{(T)} = \text{MP}_{\text{enc}}^{(T-1)}(\mathcal{G}_{\text{enc}}^{(T-1)}), \quad (1)$$
$$\mu_{z_i}, \sigma_{z_i}^2 = R_{\text{enc}}(v_{i,\text{enc}}^{(T)}).$$

In the decoder $p_\theta(\mathbf{d}|\mathbf{z}, \mathcal{G})$, each $z_i$ is concatenated with the respective node feature $v_i$ to give $v_i' \in \mathbb{R}^{F_v+1}$. Each $v_i'$ and each $e_k$ are passed to $F_{\text{dec},v}$ and $F_{\text{dec},e}$ (two MLPs), respectively, to give $\mathcal{G}_{\text{dec}}^{(0)}$, where $\mathcal{G}_{\text{dec}}^{(t)} = (\{v_{i,\text{dec}}^{(t)}\}_{i=1}^{N_v}, \{(e_{k,\text{dec}}^{(t)}, r_k, s_k)\}_{k=1}^{N_e}), v_{i,\text{dec}}^{(t)} \in \mathbb{R}^{L_v}$, and $e_{k,\text{dec}}^{(t)} \in \mathbb{R}^{L_e}$. Then, $T$ MPNNs of depth 1, $\{\text{MP}_{\text{dec}}^{(t)}\}_{t=1}^{T}$, are consecutively applied to obtain $\mathcal{G}_{\text{dec}}^{(T)}$. Finally, the read-out function $R_{\text{dec}}$ (an MLP) takes each $e_{k,\text{dec}}^{(T)}$ to predict the mean $\mu_{d_k} \in \mathbb{R}$ and the variance $\sigma_{d_k}^2 \in \mathbb{R}$ of the Gaussian distribution for $d_k$. In summary,

$$v_{i,\text{dec}}^{(0)} = F_{\text{dec},v}(v_i'), \quad e_{k,\text{dec}}^{(0)} = F_{\text{dec},e}(e_i),$$
$$\mathcal{G}_{\text{dec}}^{(1)} = \text{MP}_{\text{dec}}^{(0)}(\mathcal{G}_{\text{dec}}^{(0)}), \quad \mathcal{G}_{\text{dec}}^{(t+1)} = \text{MP}_{\text{dec}}^{(t)}(\mathcal{G}_{\text{dec}}^{(t)}), \quad \mathcal{G}_{\text{dec}}^{(T)} = \text{MP}_{\text{dec}}^{(T-1)}(\mathcal{G}_{\text{dec}}^{(T-1)}), \quad (2)$$
$$\mu_{d_k}, \sigma_{d_k}^2 = R_{\text{dec}}(e_{k,\text{dec}}^{(T)}).$$

The sets of parameters in the encoder and decoder, $\phi$ and $\theta$ (i.e., parameters in $F_{\text{enc},v}$, $F_{\text{enc},e}$, $\{\text{MP}_{\text{enc}}^{(t)}\}_{t=1}^{T}$, $R_{\text{enc}}$, $F_{\text{dec},v}$, $F_{\text{dec},e}$, $\{\text{MP}_{\text{dec}}^{(t)}\}_{t=1}^{T}$, $R_{\text{dec}}$), respectively, are optimized by maximizing the evidence lower bound (ELBO):

$$L = \mathbb{E}_{\mathbf{z} \sim q_\phi(\mathbf{z}|\mathbf{d}, \mathcal{G})}[\log p_\theta(\mathbf{d}|\mathbf{z}, \mathcal{G})] - D_{\text{KL}}[q_\phi(\mathbf{z}|\mathbf{d}, \mathcal{G})||p_\theta(\mathbf{z}|\mathcal{G})], \quad (3)$$

where the prior $p_\theta(\mathbf{z}|\mathcal{G})$ consists of factorized Gaussians. The optimal values for the hyperparameters for the network dimensions, number of message passes, batch size, and learning rate of the Adam optimizer (Kingma & Ba, 2014) were tuned by maximizing the validation performance (ELBO) with a Bayesian optimizer and are reported in Appendix A.1.3.

### 2.3 Conformation Generation through Euclidean Distance Geometry

To compute molecular properties, quantum-chemical methods need to be employed which require the input, i.e., the molecule, to be in Cartesian coordinates.[3] Therefore, we use an EDG algorithm to translate the set of distances $\{d_k\}_{k=1}^{N_e}$ to a set of atomic coordinates $\{\mathbf{r}_i\}_{i=1}^{N_v}$.[4]

EDG is the mathematical basis for a geometric theory of molecular conformation. In the field of machine learning, Weinberger & Saul (2006) used it for learning image manifolds, Tenenbaum et al. (2000) for image understanding and handwriting recognition, Jain & Saul (2004) for speech and music, and Demaine et al. (2009) for music and musical rhythms. An EDG description of a molecular system consists of a list of lower and upper bounds on the distances between pairs of atoms $\{(d_{k,\min}, d_{k,\max})\}_{k=1}^{N_e}$. Here, $p_\theta(\mathbf{d}|\mathbf{z}, \mathcal{G})$ is used to model these bounds, namely, we set the bounds to $\{(\mu_{d_k} - \sigma_{d_k}, \mu_{d_k} + \sigma_{d_k})\}$, where $\mu_{d_k}$ and $\sigma_{d_k}$ are the mean and standard deviation for each distance $d_k$ given by the CVAE. Then, an EDG algorithm determines a set of Cartesian coordinates $\{\mathbf{r}_i\}_{i=1}^{N_v}$ so that these bounds are fulfilled (see Appendix A.2 for details).[5] Together with the corresponding chemical elements $\{\epsilon_i\}_{i=1}^{N_v}$, we obtain a conformation $\mathbf{x}$.

### 2.4 Calculation of Molecular Properties

We can get an MC estimate of the expectation $\mathbb{E}_{\mathcal{G}}[\mathcal{O}]$ of a property $\mathcal{O}$ (e.g., the dipole moment) for a molecule represented by $\mathcal{G}$ by drawing conformational samples $\mathbf{x}_i \sim p(\mathbf{x}|\mathcal{G})$ and computing $\mathcal{O}(\mathbf{x}_i) \in \mathbb{R}$ with a quantum-chemical method (e.g., density functional theory). Since we cannot draw samples from $p(\mathbf{x}|\mathcal{G})$ directly, we employ an IS integration scheme (Bishop, 2009) with our CVAE as the proposal distribution. We assume that we can readily evaluate the unnormalized probability of a conformation $\tilde{p}(\mathbf{x}|\mathcal{G}) = \exp\{-E(\mathbf{x})/k_B T\}$, where $\mathbf{x}$ must be a conformation of the molecule and the energy $E(\mathbf{x})$ is determined with a quantum-chemical method. Since the EDG algorithm is mapping the distribution $p_\theta(\mathbf{d}|\mathbf{z}, \mathcal{G})$ to a point mass in $\mathbb{R}^{3N_v}$, the MC estimate for the resulting distribution $p_{\text{prop}}(\mathbf{x}|\mathcal{G})$ is given by a mixture of delta functions, each of which is centered at the $\mathbf{x}_i$ resulting from mapping $p_\theta(\mathbf{d}|\mathbf{z}_i, \mathcal{G})$ to $\mathbb{R}^{3N_v}$, where $\mathbf{z}_i \sim p_\theta(\mathbf{z}|\mathcal{G})$, that is, $p_{\text{prop}}(\mathbf{x}|\mathcal{G}) \approx \frac{1}{N} \sum_{i=1}^{N} \delta(\mathbf{x} - \mathbf{x}_i)$. The IS estimator for the expectation of $\mathcal{O}$ w. r. t. $\tilde{p}(\mathbf{x}|\mathcal{G})$ then reads

$$\hat{\mathbb{E}}_{\mathcal{G}}[\mathcal{O}] \stackrel{\text{MC}}{\approx} \frac{1}{N} \sum_{i=1}^{N} \mathcal{O}(\mathbf{x}_i) \stackrel{\text{IS}}{=} \frac{1}{N} \sum_{i=1}^{N} \mathcal{O}(\mathbf{x}_i') \frac{\tilde{p}(\mathbf{x}_i'|\mathcal{G})}{p_{\text{prop}}(\mathbf{x}_i'|\mathcal{G})}, \tag{4}$$

where $\mathbf{x}_i \sim \tilde{p}(\mathbf{x}_i|\mathcal{G})$ and $\mathbf{x}_i' \sim p_{\text{prop}}(\mathbf{x}_i'|\mathcal{G})$, so that the expectation of $\mathcal{O}$ w. r. t. the normalized version of $\tilde{p}(\mathbf{x})$ is then

$$\mathbb{E}_{\mathcal{G}}[\mathcal{O}] = \frac{\hat{\mathbb{E}}_{\mathcal{G}}[\mathcal{O}]}{\hat{\mathbb{E}}_{\mathcal{G}}[1]} \approx \frac{1}{Z} \sum_{i=1}^{N} \mathcal{O}(\mathbf{x}_i) \tilde{p}(\mathbf{x}_i'|\mathcal{G}), \tag{5}$$

where $Z \approx \sum_{i=1}^{N} \tilde{p}(\mathbf{x}_i')$ and $N$ is the number of samples. When dividing two delta functions we have assumed that they take some arbitrarily large finite value.

## 3 Related Works

The standard approach for generating molecular conformations is to start with one, and make small changes to it over time, e.g., by using MCMC or MD. These methods are considered the gold standard for sampling equilibrium states, but they are computationally expensive, especially if the molecule is large and the Hamiltonian is based on quantum-mechanical principles (Shim & MacKerell, 2011; Ballard et al., 2015; De Vivo et al., 2016).

A much faster but more approximate approach for conformation generation is EDG (Havel, 2002; Blaney & Dixon, 2007; Lagorce et al., 2009; Riniker & Landrum, 2015). Lower and upper distance

---

[3]Even though quantum-chemical methods require the input to be in Cartesian coordinates, calculated properties, such as the energy, are invariant under translation and rotation.

[4]There are additional constraints due to chirality. However, since they are given by $\mathcal{G}$ and are fixed, they are not modeled by our method.

[5]Often there exist multiple solutions for the same set of bounds. As the bounds are generally tight, the solutions are very similar. Therefore, we only generate one set of coordinates per set of bounds.

bounds for pairs of atoms in a molecule are fixed values based on ideal bond lengths, bond angles, and torsional angles. These values are often extracted from crystal structure databases (Allen, 2002). These methods aim to generate a low-energy conformation, not to generate unbiased samples from the underlying distribution at a certain temperature.

There exist several machine learning approaches as well, however, they are mostly tailored towards studying protein dynamics. For example, Noé et al. (2019) trained Boltzmann generators on the energy function of proteins to provide unbiased, one-shot samples from their equilibrium states. This is achieved by training an invertible neural network to learn a coordinate transformation from a system's configurations to a latent space representation. Further, Lemke & Peter (2019) proposed a dimensionality reduction algorithm that is based on a neural network autoencoder in combination with a nonlinear distance metric to generate samples for protein structures. Both models learn protein-specific coordinate transformations that cannot be transferred to other molecules.

AlQuraishi (2019) introduced an end-to-end differentiable recurrent geometric network for protein structure learning based on amino acid sequences. Also, Ingraham et al. (2019) proposed a neural energy simulator model for protein structure that makes use of protein sequence information. In contrast to amino acid sequences, molecular graphs are, in general, not linear but highly branched and often contain cycles. This makes them unsuitable for recurrent networks.

Finally, Mansimov et al. (2019) presented a conditional deep generative graph neural network to generate molecular conformations given a molecular graph. Their goal is to predict the most likely conformation and not a distribution over conformations. Instead of encoding molecular environments in atomic distances, they work directly in Cartesian coordinates. As a result, the generated conformations showed significant structural differences compared to the ground-truth and required refinement through a force field, which is often employed in MD simulations.

We argue that our model has several advantages over the approaches reviewed above:

- It is a fast alternative to resource-intensive approaches based on MCMC or MD.

- Our principled representation based on pair-wise distances does not restrict our approach to any particular molecular structure.

- Since our model employs message-passing neural networks, it is transferable – it can extrapolate from only a few graphs to unseen ones.

## 4 THE CONF17 BENCHMARK

The CONF17 benchmark is the first benchmark for molecular conformation sampling.[6] It is based on the ISO17 dataset (Schütt et al., 2017a) which consists of conformations of various molecules with the atomic composition $C_7H_{10}O_2$ drawn from the QM9 dataset (Ramakrishnan et al., 2014). These conformations were generated by *ab initio* molecular dynamics simulations at 500 Kelvin which generates trajectories of a single molecule covering a large variety of conformations. The CONF17 benchmark consists of 127 distinct molecular graphs each with 3380 conformations on average. We split this dataset into multiple training and test splits, each consisting of 107 and 20 graphs, respectively (see Appendix A.1 for more details).[7]

In Fig. 4, (A), the structural formulae of a random selection of molecules from this benchmark are shown. Most molecules feature highly-strained, complex 3D structures such as rings which are typical of drug-like molecules. It is thus the structural complexity of the molecules, not their number of degrees of freedom, that makes this benchmark challenging. In Fig. 4, (B), the frequency of distances (in Å) in the conformations are shown for each edge type. It can be seen that the marginal distributions of the edge distances are multimodal and highly context dependent.

---

[6]Datasets such as the one published by Kanal et al. (2018) only include *conformers*, i.e., the stable conformations of a molecule, and not a distribution over conformations.

[7]The CONF17 benchmark is publicly available online `https://figshare.com/s/1b42bf865bd78c457354`.

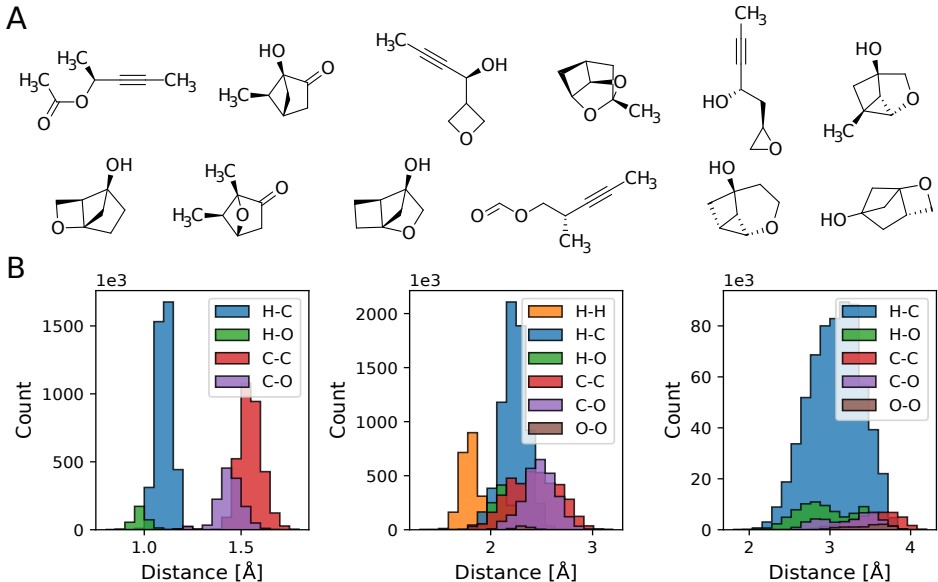

Figure 4: Overview of the CONF17 benchmark. (A) Structural formulae of a random selection of molecules. (B) Distribution of distances (in Å) grouped by edge (from left to right: $E_{\text{bond}}$, $E_{\text{angle}}$, and $E_{\text{dihedral}}$) and vertex type (chemical element).

## 5 EXPERIMENTS

We assess the performance of our method, named Graph Distance Geometry (GRAPHDG), by comparing it with two state-of-the-art methods for molecular conformation generation: RDKIT (Riniker & Landrum, 2015), a classical EDG approach, and DL4CHEM (Mansimov et al., 2019), a machine learning approach. We trained GRAPHDG and DL4CHEM on three different training and test splits of the CONF17 benchmark using Adam (Kingma & Ba, 2014). We generated 3000 conformations with each method for molecular graphs in a test set.

### 5.1 DISTRIBUTIONS OVER DISTANCES

We assessed the accuracy of the distance distributions of RDKIT, DL4CHEM, and GRAPHDG by calculating the maximum mean discrepancy (MMD) (Gretton et al., 2012) to the ground-truth distribution. We compute the MMD using a Gaussian kernel, where we set the standard deviation to be the median distance between distances $\mathbf{d}$ in the aggregate sample. For this, we determined the distances in the conformations from the ground-truth and those generated by RDKIT and DL4CHEM. For each train-test split and each $\mathcal{G}$ in a test set, we compute the MMD of the joint distribution of distances between C and O atoms (H atoms are usually ignored), the MMDs of pair-wise distances $p(d_i, d_j|\mathcal{G})$, and the MMDs between the marginals of individual distances $p(d_i|\mathcal{G})$. We aggregate the results of three train-test splits, and, finally, compute the median MMDs and average rankings. The results are summarized in Table 1. It can be seen that the samples from GRAPHDG are significantly closer to the ground-truth distribution than the other methods. RDKIT is slightly worse than GRAPHDG while DL4CHEM seems to struggle with the complexity of the molecules and the small number of graphs in the training set.

In Fig. 5, we showcase the accuracy of our model by plotting the marginal distributions $p(d_i|\mathcal{G})$ for distances between C and O atoms given a molecular graph from a test set. It can be seen that RDKIT consistently underestimates the marginal variances. This is because this method aims to predict the most stable conformation, i.e., the distribution's mode. In contrast, DL4CHEM often fails to predict the correct mean. For this molecule, GRAPHDG is the most accurate, predicting the right mean and variance in most cases. Additional figures can be found in the Appendix A.4, where we also show plots for the marginal distributions $p(d_i, d_j|\mathcal{G})$.

Table 1: Assessment of the accuracy of the distributions over conformations generated by three models compared to the ground-truth. We compare the distributions with respect to the marginals $p(d_k|\mathcal{G})$, $p(d_k, d_l|\mathcal{G})$, and the distribution over all edges between C and O atoms $p(\{d_k\}|\mathcal{G})$. Two different metrics are used: median MMD between ground-truth conformations and generated ones, and mean ranking (1 to 3) based on the MMD. Reported are the results for molecular graphs in a test set from three train-test splits. Standard errors are given in brackets.

| | Median MMD | | | Mean Ranking | | |
| | RDKIT | DL4CHEM | GRAPHDG | RDKIT | DL4CHEM | GRAPHDG |
|---|---|---|---|---|---|---|
| $p(d_k|\mathcal{G})$ | 0.55 (0.01) | 1.11 (0.01) | **0.38** (0.02) | 1.71 (0.03) | 2.74 (0.02) | **1.51** (0.03) |
| $p(d_k, d_l|\mathcal{G})$ | 0.53 (0.01) | 1.09 (0.01) | **0.34** (0.01) | 1.66 (0.02) | 2.92 (0.01) | **1.43** (0.02) |
| $p(\{d_k\}|\mathcal{G})$ | 0.60 (0.01) | 1.07 (0.03) | **0.44** (0.05) | 1.58 (0.05) | 2.90 (0.05) | **1.45** (0.02) |

Figure 5: Marginal distributions $p(d_k|\mathcal{G})$ of ground-truth and predicted bond distances (in Å) between C and O atoms given a molecular graph from the test set. The atoms connected by each edge $d_k$ are indicated in each subplot ($s_k$–$r_k$). In the 3D structure of the molecule, carbon and oxygen atoms are colored gray and red, respectively. H atoms are omitted for clarity.

## 5.2 GENERATION OF CONFORMATIONS

We passed the distances from our generative model to an EDG algorithm to obtain conformations. For 99.9% of the sets of distances, all triangle inequalities held. For 94% of the molecular graphs, the algorithm succeeded which is 8 pp higher than the success rate we observed for RDKIT. For each molecular graph in a test set, we generated 50 conformations with each method. This took DL4CHEM, RDKIT, and GRAPHDG on average around hundreds of milliseconds per molecule.[8] In contrast, a single conformation in the ISO17 dataset takes around a minute to compute. In Fig. 6, an overlay of these conformations of six molecules generated by the different methods is shown. It can be seen that RDKIT's conformations show too little variance, while DL4CHEM's structures are mostly invalid, which is due in part to its failure to predict the correct interatomic angles. Our method slightly overestimates the structural variance (see, for example, Fig. 6, top row, second column), but produces conformations that are the closest to the ground-truth.

## 5.3 CALCULATION OF MOLECULAR PROPERTIES

We estimate expected molecular properties for molecular graphs from the test set with $N = 50$ conformational samples each. Due to their poor quality, we could not compute properties $\mathcal{O}(\mathbf{x})$, including the energy $E(\mathbf{x})$, for conformations generated with DL4CHEM, and thus, this method is excluded from this analysis. In Table 2, it can be seen that RDKIT and GRAPHDG perform

---

[8] All simulations were carried out on a computer equipped with an i7-3820 CPU and a GeForce GTX 1080 Ti GPU.

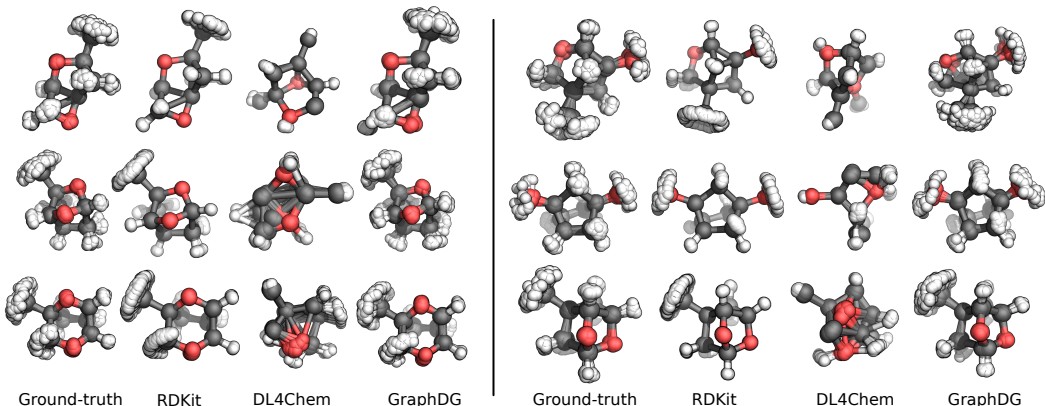

Figure 6: Overlay of 50 conformations from the ground-truth and three models based on six random molecular graphs from the test set. C, O, and H atoms are colored gray, red, and white, respectively.

Table 2: Median difference in average properties between ground-truth and RDKIT and GRAPHDG: total electronic energy $E_{\text{elec}}$ (in kJ/mol), the energy of the HOMO and the LUMO $\epsilon_{\text{LUMO}}$ and $\epsilon_{\text{LUMO}}$, respectively (in eV), and the dipole moment $\mu$ (in debye). Reported are the results for molecular graphs from the test set, averaged over three train-test splits. Standard errors are given in brackets.

|  | RDKIT | GRAPHDG |
|---|---|---|
| $E_{\text{elec}}$ | 42.7 (4.3) | 58.0 (21.0) |
| $\epsilon_{\text{HOMO}}$ | 0.08 (0.04) | 0.10 (0.05) |
| $\epsilon_{\text{LUMO}}$ | 0.15 (0.03) | 0.09 (0.05) |
| $\mu$ | 0.29 (0.05) | 0.33 (0.09) |

similarly well (see Appendix A.2 for computational details). However, both methods are still highly inaccurate for $E_{\text{elec}}$ (in practice, an accuracy of less than 5 kJ/mol is required). Close inspection of the conformations shows that, even though GRAPHDG predicts the most accurate distances overall, the variances of certain strongly constrained distances (e.g., triple bonds) are overestimated so that the energies of the conformations increase drastically.

## 6   LIMITATIONS

The first limitation of this work is that the CVAE can sample (with low probability) invalid sets of distances for which there exists no 3D structure. Second, the CONF17 benchmark covers only a small portion of chemical space. Finally, a large set of auxiliary edges would be required to capture long-range correlations (e.g., in proteins). Future work will address these points.

## 7   CONCLUSIONS

We presented GRAPHDG, a transferable, generative model that allows sampling from a distribution over molecular conformations. We developed a principled learning representation of conformations that is based on distances between atoms. Then, we proposed a challenging benchmark for comparing molecular conformation generators. With this benchmark, we show experimentally that conformations generated by GRAPHDG are closer to the ground-truth than those generated by other methods. Finally, we employ our model as a proposal distribution in an IS integration scheme to estimate molecular properties. While orbital energies and the dipole moments were predicted well, a larger and more diverse dataset will be necessary for meaningful estimates of electronic energies. Further, methods have to be devised to estimate how many conformations need to be generated to ensure all important conformations have been sampled. Finally, our model could be trained on conformational distributions at different temperatures in a transfer learning-type setting.

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

# A APPENDIX

## A.1 CONF17 BENCHMARK

### A.1.1 DATA GENERATION

The ISO17 dataset (Schütt et al., 2017a) was processed in the following way. First, conformations in which the molecular connectivity was modified (i.e., bonds were broken or new ones are formed) were discarded. For this, the tool XYZ2MOL (Jensen, 2019) was employed. Second, the molecular graphs were augmented by adding auxiliary edges for reasons described in Section 2.1. Auxiliary edges between all second neighbors were added. This can lead to a slight over-specification of the system's geometry, however, this did not pose a problem in our experiments. In addition, auxiliary edges between third neighbors were added to fix dihedral angles. Since there are potentially many ways of specifying a dihedral angle in a molecular system, we resorted to the works of Riniker & Landrum (2015) and Guba et al. (2016) to decide where to place edges between third neighbors.

### A.1.2 INPUT FEATURES

Below we list the node and edges features in the CONF17 benchmark.

Table 3: Node features.

| Feature | Data Type | Dimension |
|---|---|---|
| atomic number | integer | 1 |
| chiral tag | one-hot (R, S, and N/A) | 3 |

Table 4: Edge features.

| Feature | Data Type | Dimension |
|---|---|---|
| kind | one-hot (indicating whether $e$ is in $E_{bond}$, $E_{angle}$, or $E_{dihedral}$) | 3 |
| stereo chemistry | one-hot (E, Z, Any, None, and N/A) | 5 |
| type | integer (single, double, triple or N/A) | 1 |
| is aromatic | binary | 1 |
| is conjugated | binary | 1 |
| is in ring of size | one-hot (3, 4, ..., 9) and N/A | 8 |

### A.1.3 MODEL ARCHITECTURE

The full model is available online `https://figshare.com/s/1b42bf865bd78c457354`. In following, the hyperparameters of our model are specified:

Activations throughout this paper: ReLU; $L_v$, $L_e$: 10; $F_{enc,v}$: neural network with depth 2, width 20; $F_{enc,e}$: neural network with depth 3, width 60; $F_{dec,v}$, $F_{dec,e}$: neural networks with depth 2, width 70; $\{MP_{enc}^{(t)}\}_{t=1}^T$, $\{MP_{dec}^{(t)}\}_{t=1}^T$: MPNN width depth 1 and three multi-head attention heads, $T = 3$, for node and edge updates neural networks with depth 2, and width 70 were used. $R_{enc}$, $R_{dec}$: neural networks with depth 2, width 70. Batch size: 16 (conformations);

## A.2 COMPUTATIONAL DETAILS

### A.2.1 QUANTUM-CHEMICAL CALCULATIONS

All quantum-chemical calculations were carried out with the `PySCF` program package (version 1.5) (Sun et al., 2018) employing the exchange-correlation density functional PBE (Perdew et al., 1996), and the def2-SVP (Weigend & Ahlrichs, 2005; Weigend, 2006) basis set.

Conformations generated by DL4CHEM did not succeed as some atoms were too close to each other. Self-consistent field algorithms in quantum-chemical software such as `PySCF` do not converge for such molecular structures.

With quantum-chemical methods, we calculate several properties that concern the states of the electrons in the conformation. These are the total electronic energy $E_{\text{elec}}$, the energy of the electron in the highest occupied molecular orbital (HOMO in eV) $\epsilon_{\text{HOMO}}$, the energy of the lowest unoccupied molecular orbital (LUMO in eV) $\epsilon_{\text{LUMO}}$, and the norm of the dipole moment $\mu$ (in debye).

### A.2.2 EUCLIDEAN DISTANCE GEOMETRY

We refer the reader to Havel (2002) for theory on EDG, algorithms, and chemical applications. In summary, the EDG procedure consists of the following three steps:

1. Bound smoothing: extrapolating a complete set of lower and upper limits on all the distances from the sparse set of lower and upper bounds.

2. Embedding: choosing a random distance matrix from within these limits, and computing coordinates that are a certain best-fit to the distances.

3. Optimization: optimizing these coordinates versus an error function which measures the total violation of the distance (and chirality) constraints.

We use the EDG implementation found in RDKIT (Riniker & Landrum, 2015) with default settings.

### A.3 GENERATION OF CONFORMATIONS

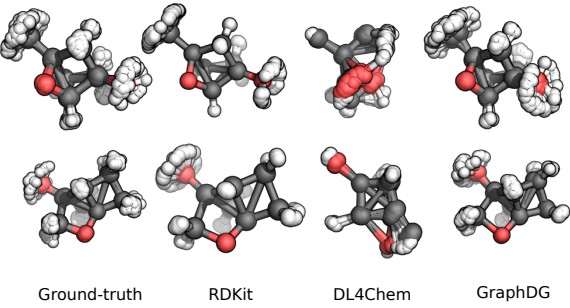

Figure 7: Overlay of 50 conformations from the ground-truth, RDKIT, DL4CHEM, and GRAPHDG based on two random molecular graphs from the test set. C, O, and H atoms are colored gray, red, and white, respectively.

### A.4 DISTRIBUTIONS OVER DISTANCES

Below, the marginal distributions of the distances for a variety of molecular graphs are shown.

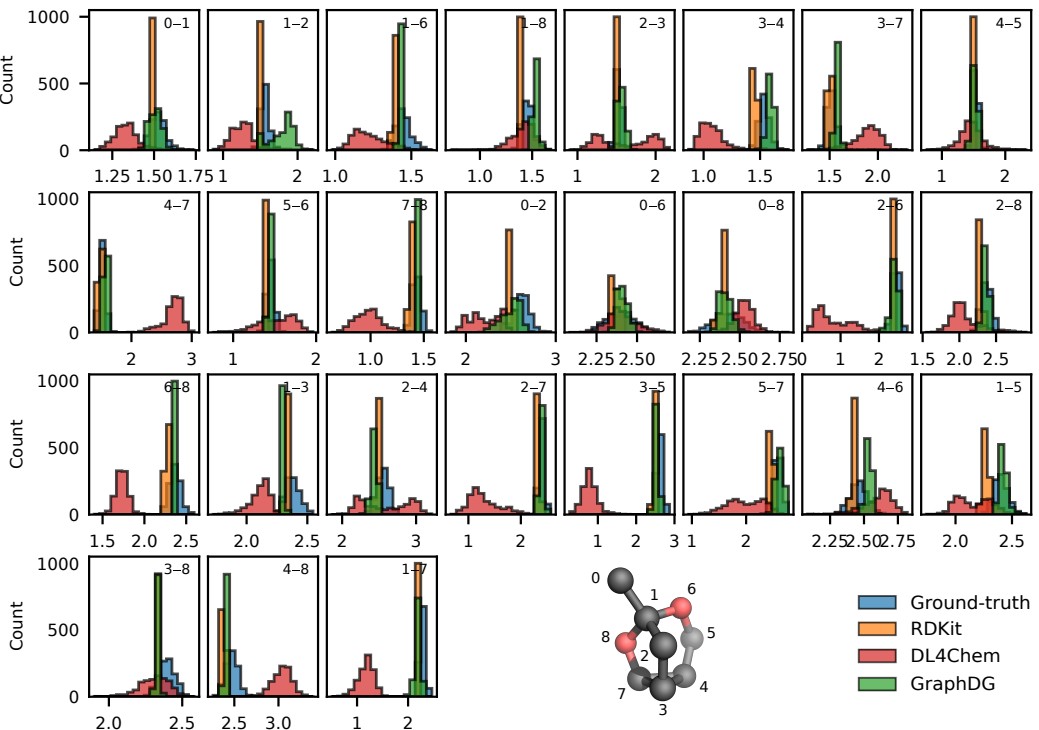

Figure 8: Marginal distributions $p(d_k|\mathcal{G})$ of ground-truth and predicted distances (in Å) between C and O atoms given a molecular graph from the test set. The atoms connected by each edge $d_k$ are indicated in each subplot $(s_k-r_k)$. In the 3D structure of the molecule, carbon and oxygen atoms are colored gray and red, respectively. H atoms are omitted for clarity.

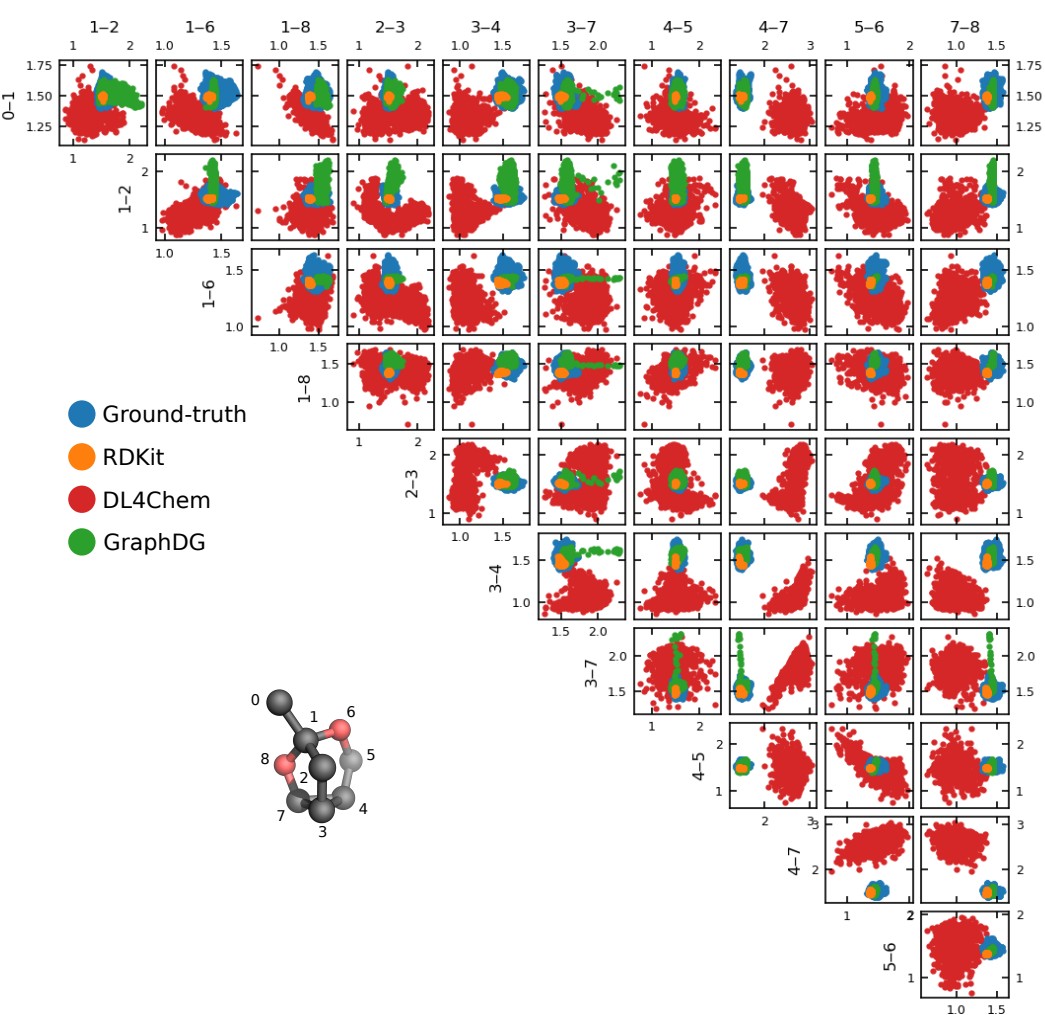

Figure 9: Marginal distributions $p(d_i, d_j | \mathcal{G})$ of ground-truth and predicted distances for a molecular graph from the test set (in Å). Here, $d_i$ and $d_j$ are restricted to edges representing bonds between C and O atoms. In the 3D structure of the molecule, carbon and oxygen atoms are colored gray and red, respectively. H atoms are omitted for clarity.

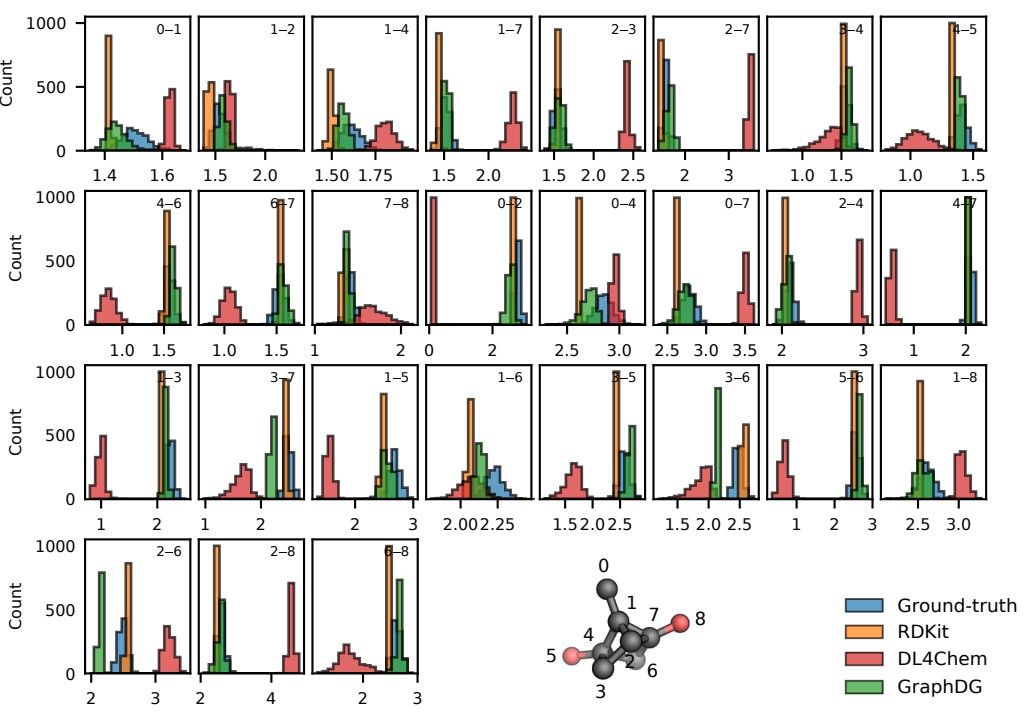

Figure 10: See caption of Fig. 8

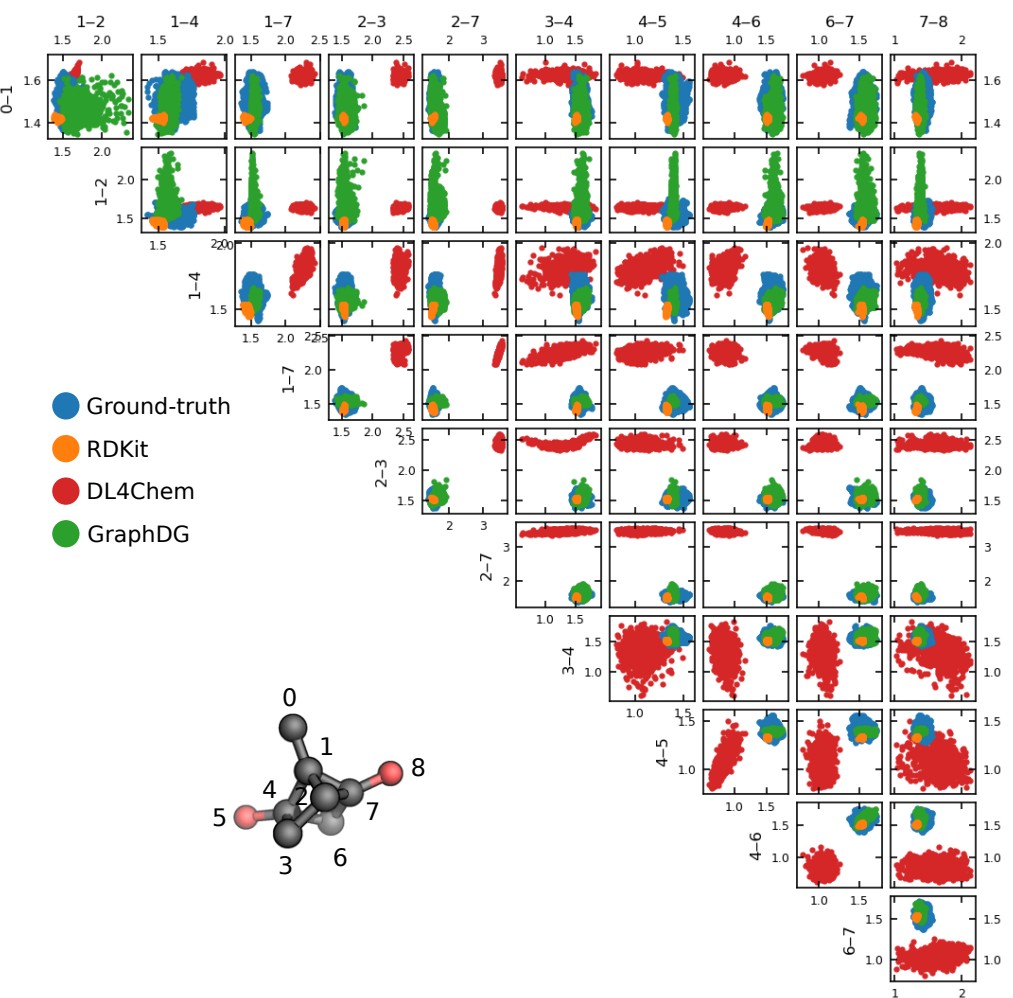

Figure 11: See caption of Fig. 9

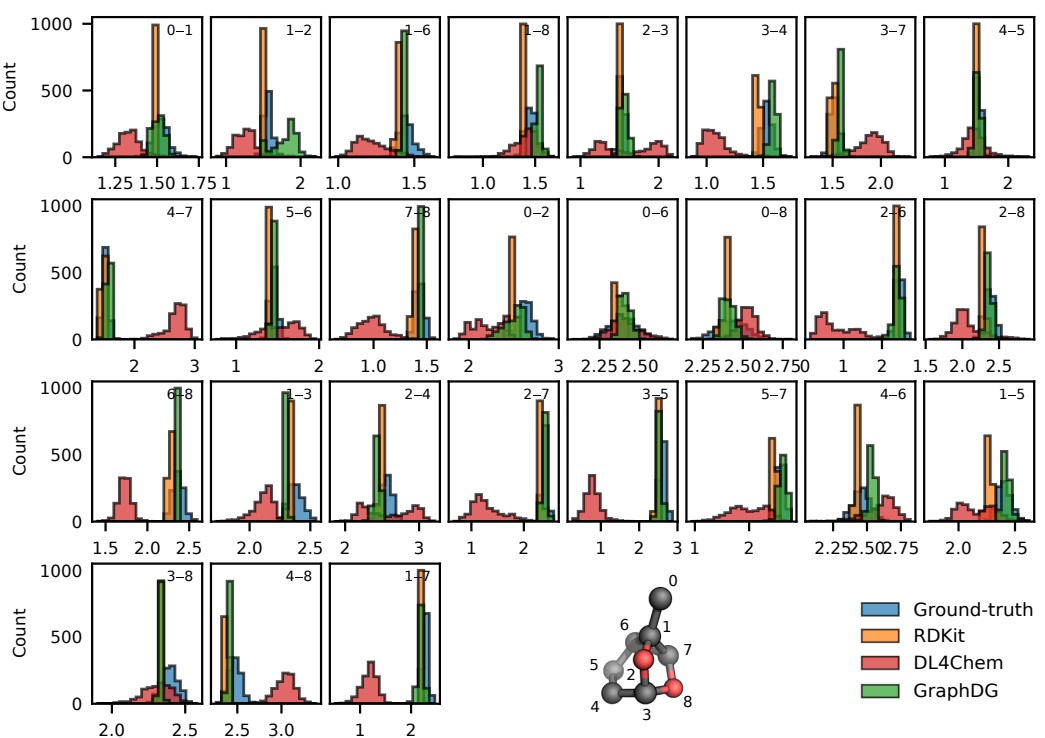

Figure 12: See caption of Fig. 8

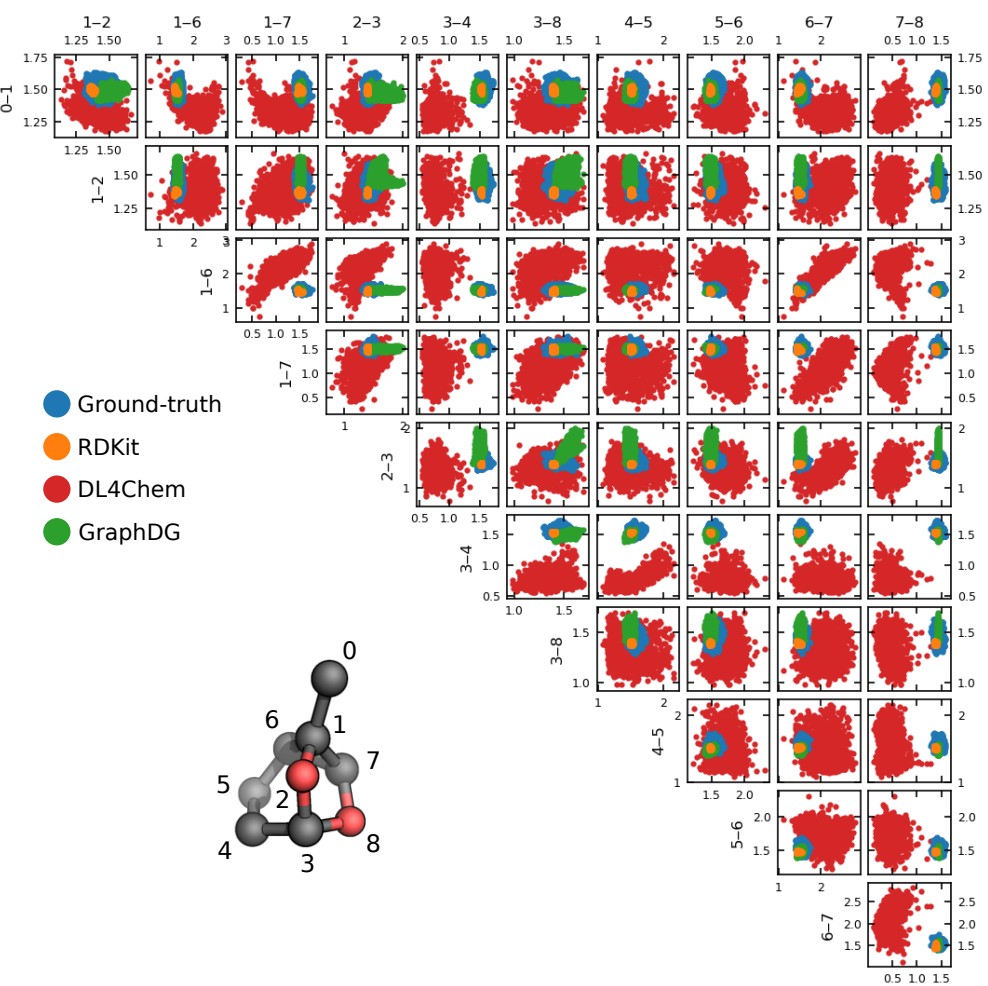

Figure 13: See caption of Fig. 9

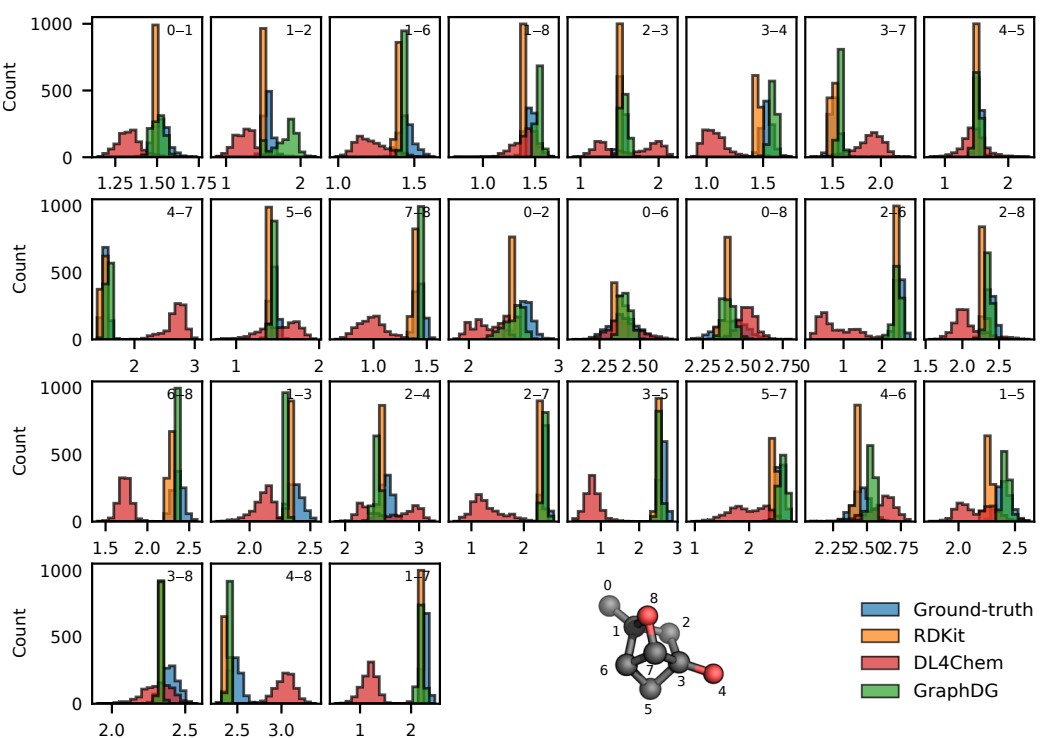

Figure 14: See caption of Fig. 8

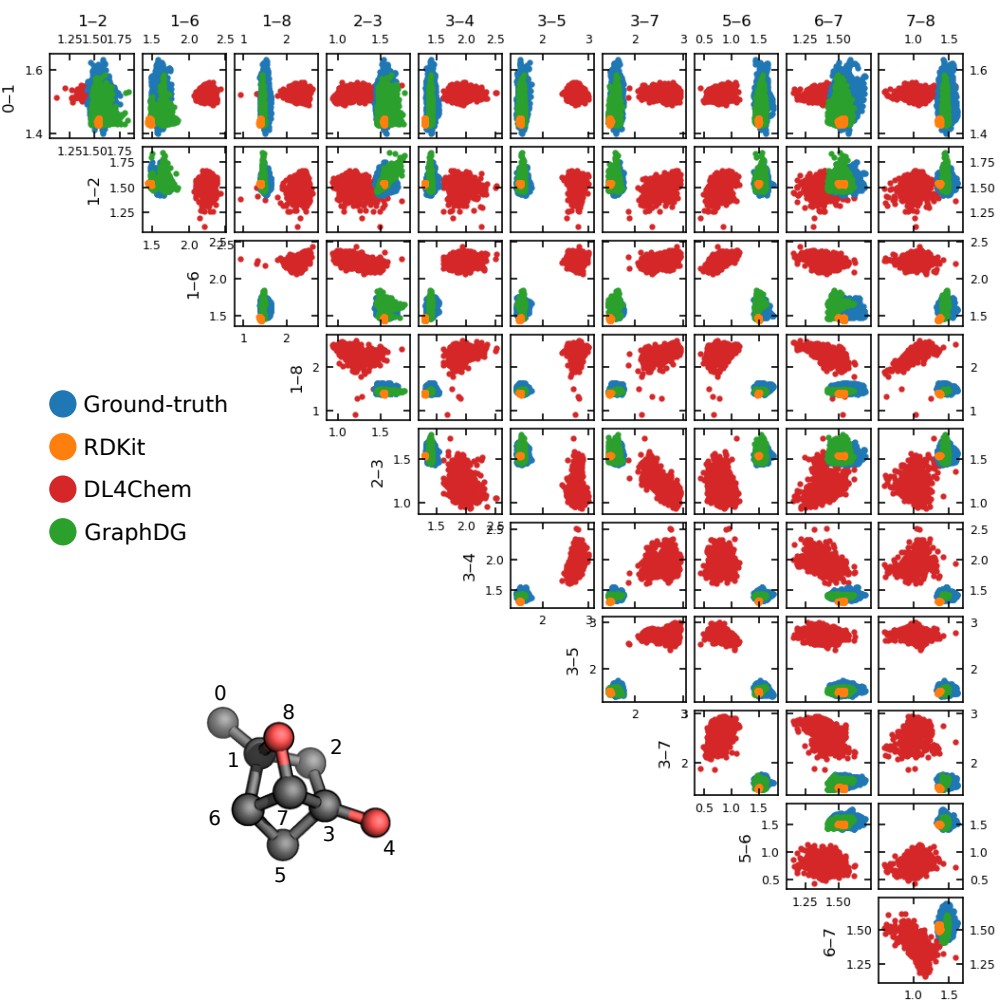

Figure 15: See caption of Fig. 9

