# OpenReview forum: "A Generative Model for Molecular Distance Geometry"
_ICLR.cc/2020/Conference — Reject_

### Official Review · AnonReviewer3 · 2019-10-19
**Official Blind Review #3**

**Rating:** 3

**Review:**


In this paper, a new method for estimating the equilibrium measure of molecular compounds is proposed and tested on a new (enriched) version of an existing data set (CONF17 is derived from the pre-existing ISO17, itself extracted from QM9).
The method takes as input a collection of graph structures of a given stoichiometry mixture, and learns the corresponding conformations.
This results in a generative model, taking as input a graph structure (or actually a single conformation instance, thus setting the stoichiometry AND the isomer/graph structure), to generate (in a one-shot fashion) other conformations of that isomer.
The distribution of conformations sampled in this way compare well with those of an extensive training set (CONF17, set of isomers and conformations of a single compound), and in particular they perform better than 2 alternative methods published recently (2015 and 2019, the former performing globally much better than the latter!).


Concretely, the model/method/algorithm consists in a combination of existing network architectures.  The main original contribution is to map the Cartesian coordinates of atoms into a pairwise distance space, which is naturally translationally and rotationally invariant.  Some of the second and third neighbors distances in the graph structure are considered, such that the conformation is completely set (this would not be the case if considering only 1st nearest neighbors distances).
The precise combination of architectures is explained in section 2.2. This explanation is a little fast for me (but I am new to this field).
In that section, it could be nice to precise that the job of the CVAE is to learn (among others) the parameters mu,sigma of q_phi(z|d,G), using a batch of conformations of a given isomer (and then generalize to other isomers, each time guessing the correct mu,sigma parameters, even for a new isomer).
This is not obvious for the inexperienced reader (I hope I "guessed" correctly).

The "new dataset" shared in this work is indeed useful for reproducing results and for later comparison with other works which aim at sampling conformations.
However, it derives directly from ISO17, which is itself an extraction from QM9. ISO17 lists the Cartesian coordinates of 127 isomers of C7,H10,O2 (each having ~3000 conformations).  The "new" CONF17 is novel only in that it provides these conformations in distance-matrix space, i.e. it sets which pairs of atoms need to have their distances computed.  This is very useful because this choice of pairs is not unique, and so setting it once is necessary for comparison.  I would not stress too much that this is a new dataset: at first it actually sounds like authors chose a dataset such that their method outperforms others'!  Which turns out not to be the case (they chose C7,H10,O2, but this doesn't look like fine-tuning).  Indeed, their sharing of this augmented version of ISO17 is absolutely necessary: I would simply put it differently, stating from the beginning that they share an augmented version of a pre-existing data set.

About the experimental protocol/methodology:
the work carried out seems quite impressive. Indeed, the paper presents a rather extensive comparison of the new model with 2 other ones, on a large data set.
However sometimes the presentation rushes a bit towards results, in the sense that it is not easy to figure out over which ensemble the mean/median/max/min has been computed. This is related to the intrinsic complexity of estimating accuracy, given the encapsulation (compound-> isomer-> conformations-> each bond-> distribution over bond properties).  I think however it could be made clearer, and this would benefit the paper greatly.
A deeper comment: in the field of conformation sampling, there are two issues. One is to correctly sample states in each local minimum of the free energy. This is addressed in the paper. The second issue is to hunt for all possible local minima of the isomer, i.e. sample even unlikely but crucial conformations. This is not addressed here.  For computing average energy, dipole moment etc, the most likely conformations will dominate, most of the time.  However given the topic I think it would be fair to mention this point, and explain why in this specific case, the second issue is not crucial.
Because of this, also, I think

Overall, the paper is well written and is very well situated within the literature (including the very recent literature).
It states clearly its contributions and results, and provides detailed presentation of experimental results.
I think it contributes to the field and provides an interesting new way of dealing with the problem of molecular conformations sampling.  The results obtained are much better than those of the compared ML method (RDkit).
However, some claims (in introduction or in section 3) are stated more strongly than they are supported by direct experimental evidence.
- ''... the first work on sampling molecular conformations for molecules of arbitrary size and shape based on deep learning.''
However, the experiments have been performed only on isomers of  C7,H10,O2.
Furthermore, there is no discussion on the computational cost of the method at all, so its extendibility to larger shapes is not obvious.
- '' We create a new, challenging benchmark dataset CONF 17 for conformation generation, which is made publicly available.''
I made remarks on this claim above.''
- '' We develop a rigorous experimental approach for evaluating and comparing the accuracy of conformation generation methods based on the mean maximum deviation distance metric.''
However the paper's methodology doesn't address the issue of ''hunting for new local minima''.
The data set could be enriched also with a labeling of ''which conformation is in which local minimum of free energy''
- ''It is a fast alternative to resource-intensive approaches based on MCMC or MD.''
As there was no mention of speed of computation in the paper (although we can guess it is faster), this is unsupported. However it is easy to remedy, simply give an order of magnitude of sampling time (for thousands of conformations of 1 isomer).
- ''Our principled representation based on pair-wise distances does not restrict our approach to any type of system (e.g., proteins) or any particular graph structure. In addition, it is extendable in a systematic fashion.''
This suggests that the method would also work for dense packings. However this is not shown in the experiments.
Probably I misunderstood the claim, and it only refers to the ability to tackle general molecular compounds, not just linear ones?  However proteins are not always linear, but can present cycles, and other branch-style patterns, so the claim is not very clear to me.

Because some of the important claims are only weakly supported, and because of the relatively small novelty of the method, I lean on rejecting the paper.



Additional comments (for improving the paper):

There is no discussion of hyper-parameter tuning, or any detail of the intermediate GNN that constitute the encoder and decoder in the CVAE.
So it was either not done at all, and then it should be mentioned; or it was done and it should be discussed quickly.
In particular, in section 2.2 there is no discussion on the impact of the latent space' dimension, here set to Nv, i.e. 7+2+10=19 (correct?).  I guess there is a balance between accuracy and generalizability, a short discussion on this point would be interesting.

The paper should clarify the hierarchy of molecular configurations:
for a given composition (%C, %O, %H, etc) (here, fixed throughout the paper at C7,H10,O2)
there are several structures possible (graphs or isomers),
for a given graph (isomer), there can be a couple of chiral variants (mirroring of (pieces of) the molecule,
for each chiral variant, there are conformations, which correspond to rather weak variations of the Cartesian coordinates, due to thermal fluctuations. Some of these variations can be important, e.g. when a bond can freely rotate. Some correspond to bending or torsion, etc, of the molecule.
The algorithm presented in the paper attempts to sample appropriately the equilibrium distribution of conformations for each given graph structure (at fixed chirality, if I understood correctly).

Add a comment on why (in your opinion) the distributions are not exactly correctly sampled, which after all is the goal (since the initial graph structure is given as input here).


Additional question, for curiosity. In section 2.3: only a single molecule output is taken from each distance matrix.
Could you comment on the possibility to sample several Cartesian coordinates, for each distance matrix ?  It seems like several solutions should (often) exist.
Sampling continuously from that distribution would surely provide a more continuous (rich) output (am I correct) ?


**Experience Assessment:**

I do not know much about this area.

**Review Assessment: Checking Correctness Of Derivations And Theory:**

N/A

**Review Assessment: Checking Correctness Of Experiments:**

I assessed the sensibility of the experiments.

**Review Assessment: Thoroughness In Paper Reading:**

I read the paper thoroughly.

---

> ### Author Response · Authors · 2019-11-11
> **Response to Review #3**
>
> We would like to thank the reviewer for this extensive review and helpful feedback. We have incorporated all the reviewer's remarks and suggestions into the latest version of the manuscript. We also followed his advice on how to further improve the paper by running additional experiments.
>
> General Comments:
> ================
>
> We would like to thank all the reviewers for their thorough and constructive feedback.
> We made the following major changes to the paper based on the comments from all reviewers:
>
> 1. We improved the overall readability of the paper, especially for readers who are not familiar with the field of molecular modeling.
>   - We reworked the introduction to clarify the concept of a conformation.
>   - We replaced Fig. 1 in the introduction with a new figure that clearly shows how a molecule's conformations are related to its graph representation.
>
> 2. In Section 2.1, we expanded on the explanation of the molecular graph and the distances between atoms in a conformation.
>   - We now define all node and edge features formally.
>   - We modified Fig. 2 to illustrate how the set of distances $\mathbf{d}$ are extracted from the conformation $\mathbf{x}$.
>
> 3. In Section 2.2, we significantly improved the clarity of the explanation of the generative model.
>   - With additional equations and implementation details, the reader is now guided through the model and the overall data flow step by step.
>   - We added a new figure (Fig. 3) that shows the different components of the variational autoencoder and how the graph data flows through the model.
>   - We added additional information on how message-passing neural networks take graphs of arbitrary size and shape as input.
>   - All hyperparameters are now reported in the Appendix.
>
> 4. We simplified Section 2.4 on the proposed importance sampling scheme to improve readability and clarity.
>
> 5. We now propose a new benchmark (Conf17) instead of a new dataset.
>
> 6. We ran additional experiments:
>   - We studied the effect of the latent space size and report our results.
>   - We ran simulations to add information on the computational effort of all methods.
>
> Individual Response:
> ====================
>
> 1. "The precise combination of architectures is explained in section 2.2. This explanation is a little fast for me (but I am new to this field). In that section, it could be nice to precise that the job of the CVAE is to learn (among others) the parameters mu,sigma of q_phi(z|d,G), using a batch of conformations of a given isomer (and then generalize to other isomers, each time guessing the correct mu,sigma parameters, even for a new isomer). This is not obvious for the inexperienced reader (I hope I "guessed" correctly)."
>
> We acknowledge that the model definition in Section 2 was not detailed enough, and therefore, completely reworked this section. In the latest version of the manuscript, we explicitly state the role (and dimensionality) of every function and variable that appears in the implementation of the CVAE. Further, we added equations to define more formally every part of the model. Further, we replaced Fig. 2 B) with an illustration in which we show how the distances are extracted from the conformations. Finally, we added another figure (Fig. 3) that shows exactly how the graph data is flowing through the model. We hope that it is more apparent now how the CVAE learns parameters mu and sigma of q_phi(z|d,G).
>
>
> 2. "Indeed, their sharing of this augmented version of ISO17 is absolutely necessary: I would simply put it differently, stating from the beginning that they share an augmented version of a pre-existing data set."
>
> We thank you for this constructive feedback on this matter. We agree with you and now stress that this is not a new dataset but a benchmark using an existing dataset and rephrased this matter throughout our manuscript.
>
> In the introduction of the latest version of the paper it now reads:
> "We create a new, challenging benchmark for conformation generation, which is made publicly available."
>
> In the spirit of reproducibility and since arriving at the data required for running this benchmark is non-trivial, we decided to publish the data we employed in this study. Further, we updated the presentation of this benchmark in Section 4 accordingly to reflect that.

---

> > ### Author Response · Authors · 2019-11-11
> > **Response to Review #3 - continued**
> >
> > 3. "However sometimes the presentation rushes a bit towards results, in the sense that it is not easy to figure out over which ensemble the mean/median/max/min has been computed."
> >
> > We expanded on the experimental setup in Section 5.1 to address this issue and hope that it is clear now how we arrived at our results.
> >
> > 4. "A deeper comment: in the field of conformation sampling, there are two issues. One is to correctly sample states in each local minimum of the free energy. This is addressed in the paper. The second issue is to hunt for all possible local minima of the isomer, i.e. sample even unlikely but crucial conformations. This is not addressed here. ..."
> >
> > We highly appreciate your general comment on this topic. We agree that hunting for local minima is an outstanding challenge in conformational sampling. While many advanced techniques (e.g., umbrella sampling) exist, none of them can assure that all local minima have been sampled sufficiently for complex molecules. For larger molecules, it is practically impossible to guarantee that all local minima have been visited at least once. All methods we compare our method with, including RDKit and the simulation used to generate the ISO17 dataset, suffer from this shortcoming.
> >
> > However, the aim of this method is not to only generate conformations in local minima but to sample from the distribution $p(\mathbf{x} | \mathcal{G})$ in a one-shot fashion. While minimum-energy conformations will be sampled more frequently than out-of-equilibrium conformations our model is not restricted to those. Similarly, an MD or MCMC simulation does not only visit minimum energy conformations. However, we agree that minimum-energy conformations strongly contribute to average properties (e.g., dipole moment). Nonetheless, as can be seen from the results in Table 2, even highly optimized algorithms like the one in RDKit struggle to correctly sample local minima of highly constrained molecules.
> >
> > To clarify this matter, we wrote in the benchmark section:
> > "Most molecules feature highly-strained, complex 3D structures such as rings which are typical of drug-like molecules. It is thus the structural complexity of the molecules, not their number of degrees of freedom, that makes this benchmark challenging."
> >
> > Further, we now address that issue by writing in the Conclusions:
> > "Further, methods have to be devised to estimate how many conformations need to be generated to ensure all important conformations have been sampled."
> >
> > Finally, we stress the important distinction between conformers (minimum-energy structures) and conformations:
> > "Datasets such as the one published by Kanal et al. only include conformers, i.e., the most stable conformations of a molecule, and not a distribution over conformations."
> >
> >
> > 5. "However, the experiments have been performed only on isomers of  C7,H10,O2.
> > Furthermore, there is no discussion on the computational cost of the method at all, so its extendibility to larger shapes is not obvious."
> >
> > We clarify in the model definition why our method can be applied to molecules of arbitrary shape:
> > "At the center of this model are message-passing neural networks (MPNNs) (Gilmer 2017) with multi-head attention (Velickovic 2018). In short, an MPNN is a convolutional neural network that allows end-to-end learning of prediction pipelines whose inputs are graphs of arbitrary size and shape. In a convolution, neighboring nodes exchange so-called messages between neighbors to update their attributes. Edges update their attributes with the features of the nodes they are connecting. It is a well-studied technique that achieves state-of-the-art performance in representation learning for molecules."
> >
> > While it is true that the benchmark only comprises molecules with the C7H10O2 stoichiometry, MPNNs have been reported to work very well for multiple regression task on different-sized molecules in the original publication by Duvenaud et al. Therefore, we have reasons to believe that this should hold for this task as well.
> >
> >
> > 6. We addressed your remarks regarding claim 3 (the dataset / benchmark) in our answer Nr. 2.
> >
> >
> > 7. "However the paper's methodology doesn't address the issue of ''hunting for new local minima''.
> >
> > In response Nr. 4, we discuss this issue of local minima.
> >
> >
> > 8. "The data set could be enriched also with a labeling of ''which conformation is in which local minimum of free energy''."
> >
> > The ISO17 dataset was generated by an MD simulation at 500 K. Therefore, it is a difficult task to assign a snapshot of this simulation unambiguously to a particular minimum-energy conformation, especially when one considers the simulation temperature.

---

> > > ### Author Response · Authors · 2019-11-11
> > > **Response to Review #3 - continued 2**
> > >
> > > 9. "As there was no mention of speed of computation in the paper (although we can guess it is faster), this is unsupported. However it is easy to remedy, simply give an order of magnitude of sampling time (for thousands of conformations of 1 isomer)."
> > >
> > > Based on this comment we ran some additional experiments. In the results section we now write:
> > > "For each molecule in the test set, we generated 50 conformations with each method. This took DL4Chem, RDKit, and GraphDG on average around hundreds of milliseconds per molecule. In contrast, a single conformation in the ISO17 dataset takes around a minute to compute."
> > >
> > > We also state the hardware configuration of the computer these computations were carried out on.
> > > "All simulations were carried out on a computer equipped with an i7-3820 CPU and a GeForce GTX 1080 Ti GPU."
> > >
> > >
> > > 10. "This suggests that the method would also work for dense packings. However this is not shown in the experiments.
> > > Probably I misunderstood the claim, and it only refers to the ability to tackle general molecular compounds, not just linear ones?  However proteins are not always linear, but can present cycles, and other branch-style patterns, so the claim is not very clear to me."
> > >
> > > We thank the reviewer for reading our submission so carefully. We rephrased our statement to eliminate that confusion:
> > > "Our principled representation based on pair-wise distances does not restrict our approach to any particular molecular structure."
> > >
> > > With this statement, we aimed to highlight that many previous approaches are heavily tailored towards proteins (as they rely on their [linear] amino acid sequence) and cannot be directly applied to molecules that often feature cycles and many branches.
> > >
> > >
> > > 11. "There is no discussion of hyper-parameter tuning, or any detail of the intermediate GNN that constitute the encoder and decoder in the CVAE. So it was either not done at all, and then it should be mentioned; or it was done and it should be discussed quickly."
> > >
> > > In the latest version of the manuscript, we detail how the individual hyperparameters were tuned with a Bayesian optimizer. In the Appendix, we list all hyperparameters and the values that were identified by the Bayesian optimization procedure.
> > >
> > >
> > > 12. "In particular, in section 2.2 there is no discussion on the impact of the latent space' dimension, here set to Nv, i.e. 7+2+10=19 (correct?).  I guess there is a balance between accuracy and generalizability, a short discussion on this point would be interesting."
> > >
> > > Yes, that is correct, the latent space size is 19. We ran multiple experiments with latent space sizes of Nv, 2 Nv, and 3 Nv to study the effect of the latent space size
> > > on the model's performance. Our experiments showed that a latent space of 2 Nv performs slightly worse than Nv. 3 Nv results in overfitting.
> > >
> > > We now mention our results and justify our choice:
> > > "A conformation has, in general, $3N_v - 6$ spatial degrees of freedom (dofs): one dof per spacial dimension per atom minus three translational and three rotational dofs. Therefore, the latent space should be proportional to the number of atoms in the molecule. In addition, the latent space should be smaller than $3N_v$ as it is the role of the encoder to project the conformation into a lower-dimensional space.
> > > As a result, we set $k=1$ to avoid overfitting."
> > >
> > > In a footnote, we then write:
> > > "Experiments showed that our model performs similarly with a latent space of R^{2N_v} and overfits with latent spaces of R^{3N_v}."

---

> > > > ### Author Response · Authors · 2019-11-11
> > > > **Response to Review #3 - continued 3**
> > > >
> > > > 13. "The paper should clarify the hierarchy of molecular configurations:
> > > > for a given composition (%C, %O, %H, etc) (here, fixed throughout the paper at C7,H10,O2)
> > > > there are several structures possible (graphs or isomers),
> > > > for a given graph (isomer), there can be a couple of chiral variants (mirroring of (pieces of) the molecule,
> > > > for each chiral variant, there are conformations, which correspond to rather weak variations of the Cartesian coordinates, due to thermal fluctuations. Some of these variations can be important, e.g. when a bond can freely rotate.
> > > > Some correspond to bending or torsion, etc, of the molecule."
> > > >
> > > > Based on this suggestion, we rewrote the relevant section in the introduction.
> > > > Over the last few years, many highly-effective deep learning methods generating small molecules with desired properties (e.g., novel drugs) have emerged. These methods operate using graph representations of molecules in which nodes and edges represent atoms and bonds, respectively. A representation that is closer to the physical system is one in which a molecule is described by its geometry or conformation. A conformation $\mathbf{x}$ of a molecule is defined by a set of atoms $\{ (\epsilon_i, \mathbf{r}_i)\}_{i=1}^{N_v}$, where $N_v$ is the number of atoms in the molecule, $\epsilon_i \in \{\text{H}, \text{C}, \text{O},... \}$ is the chemical element of the atom $i$, and $\mathbf{r}_i \in \mathbb{R}^3$ is its position in Cartesian coordinates. Importantly, the relative positions of the atoms are restricted by the bonds in the molecule and the angles between them. Due to thermal fluctuations resulting in stretching of and rotations around bonds, there exist infinitely many conformations of a molecule. A molecule's graph representation and a set of its conformations are shown in Fig. 1.
> > > >
> > > > However, we would like to say that we don't explicitly state the relation between composition and conformations as it could confuse the reader. In addition, the concept of the composition is not strictly necessary to understand how our model works.  Furthermore, the model is (as explained above) transferable to different stoichiometries. Nonetheless, and as mentioned before, we replaced Fig. 1 to clarify how conformations and structural formulae relate.
> > > >
> > > >
> > > > 14. "Add a comment on why (in your opinion) the distributions are not exactly correctly sampled, which after all is the goal (since the initial graph structure is given as input here)."
> > > >
> > > > The reason why the distributions are not sampled exactly is that we only report performance metrics on the test set, i.e., graphs the model has not seen before. Since the benchmark is challenging (e.g., it contains many constrained rings), our model does not always predict the right distances. Nonetheless, it does significantly better than previous methods (see, for example, Table 1).
> > > >
> > > > In the manuscript, we now stress that the results are for graphs from the test set only.
> > > > "For each train-test split and each molecule G in the test set, we compute the MMD of the joint distribution of distances between C and O atoms (H atoms are usually ignored)..."
> > > >
> > > > Further, in the caption of Table 1 it now says:
> > > > "Reported are the results for molecules in the test set from three train-test splits."
> > > >
> > > >
> > > > 15. "In section 2.3: only a single molecule output is taken from each distance matrix.
> > > > Could you comment on the possibility to sample several Cartesian coordinates, for each distance matrix ?  It seems like several solutions should (often) exist. Sampling continuously from that distribution would surely provide a more continuous (rich) output (am I correct)?"
> > > >
> > > > Indeed, there exist (in general) multiple solutions for the same distance matrix. However, since the upper and lower bounds (d_min, d_max) are pretty close together, the different solutions are very similar. Therefore, we decided to take only one set of coordinates per set of bounds.
> > > >
> > > > In the latest version of the manuscript, we now write in a footnote:
> > > > "Often there exist multiple solutions for the same set of bounds. As the bounds are generally tight, the solutions are very similar. Therefore, we only generate one set of coordinates per set of bounds."
> > > >
> > > >
> > > > Again, thank you very much for your extensive review. We would appreciate it if you could let us know if you are willing to accept the paper now. If not, could you please let us know what we need to clarify further.

---

### Official Review · AnonReviewer2 · 2019-10-23
**Official Blind Review #2**

**Rating:** 6

**Review:**

Summary:
The authors propose a generative model designed for molecules, which is essentially a conditional variational auto-encoder. The model learns to generate, when conditioning on a molecule graph, the distribution of distances between each of the atoms and its second and third neighbour. Finally, using these information new molecules can be generated, which satisfy the generated distances. In the experiments the authors show the effectiveness of the method.

General Comments:
The paper is ok written, but as a non-expert of the field, I find it quite hard to follow. More specifically, I think that the main content aims to readers which are very familiar with the problem of molecule modeling, which is of course understandable. However, I find the technical (machine learning) content a bit unclear, because I think that the proposed method is not presented properly in details (see comments):

1. I think that authors should specify the dimensionalities of each variable and function in the paper. Unfortunately, none of them is clearly defined, which makes the technical part a bit hard to follow.

2. As a non-expert of the filed, I would like to know what exactly is a conformation x? Does it refer to the Euclidean coordinates of each node (atom) v of the graph? (Related to comment #1.)

3. For each G, there are multiple possible conformations x_i, where each x_i specifies the Euclidean coordinates of all the nodes (atoms) v_j? Also, each node v_j has some attributes? So the goal is to represent each conformation x_i -implicitly- as a set of distances between the nodes v_j? I think that Sec 2.1. can be more clear about the definition of the data, such that to be accessible from non experts in molecule modeling. (Related to comment #1.)

4. I think that the definition of the model in Sec 2.2. could have been better. As before, I believe that it is hard to understand explicitly what the functions input and output is, since the dimensionalities are not defined. In my opinion, it would have been better if instead of this high level text you could provide explicitly all the steps of the model e.g. the concatenations, the appends, etc.

5. I could not understand if and how the model is able to handle graphs with different number of nodes, and different number of distances among them.

6. The experimental section seems solid enough, and shows that the proposed approach works better than the other methods. Also, I like the fact that some of the limitations of the method are stated. Moreover, the related work seems to be properly included. However, since I am not an expert of the field, I am not able to provide precise feedback for these two parts.

In general, I think that the proposed model solves good enough the problem that is designed for. Also, in my opinion, the manuscript aims for specialized readers, which is definitely understandable. However, my main concern regarding the current version, is that the technical (machine learning) content is a bit unclear and probably too high level. In case the technical content was presented properly, I think that it would have been a much better fit for this community.

**Experience Assessment:**

I do not know much about this area.

**Review Assessment: Checking Correctness Of Derivations And Theory:**

I did not assess the derivations or theory.

**Review Assessment: Checking Correctness Of Experiments:**

I did not assess the experiments.

**Review Assessment: Thoroughness In Paper Reading:**

I made a quick assessment of this paper.

---

> ### Author Response · Authors · 2019-11-11
> **Response to Review #2**
>
> We thank you for your extensive review, list of comments, and constructive feedback. We have incorporated all your remarks in the latest revision of the manuscript.
>
> General Response:
> ================
>
> We made the following major changes to the paper based on the comments from all reviewers:
>
> 1. We improved the overall readability of the paper, especially for readers who are not familiar with the field of molecular modeling.
>   - We reworked the introduction to clarify the concept of a conformation.
>   - We replaced Fig. 1 in the introduction with a new figure that clearly shows how a molecule's conformations are related to its graph representation.
>
> 2. In Section 2.1, we expanded on the explanation of the molecular graph and the distances between atoms in a conformation.
>   - We now define all node and edge features formally.
>   - We modified Fig. 2 to illustrate how the set of distances $\mathbf{d}$ are extracted from the conformation $\mathbf{x}$.
>
> 3. In Section 2.2, we significantly improved the clarity of the explanation of the generative model.
>   - With additional equations and implementation details, the reader is now guided through the model and the overall data flow step by step.
>   - We added a new figure (Fig. 3) that shows the different components of the variational autoencoder and how the graph data flows through the model.
>   - We added additional information on how message-passing neural networks take graphs of arbitrary size and shape as input.
>   - All hyperparameters are now reported in the Appendix.
>
> 4. We simplified Section 2.4 on the proposed importance sampling scheme to improve readability and clarity.
>
> 5. We now propose a new benchmark (Conf17) instead of a new dataset.
>
> 6. We ran additional experiments:
>   - We studied the effect of the latent space size and report our results.
>   - We ran simulations to add information on the computational effort of all methods.
>
>
> Individual Response:
> ==================
>
> 1. "I think that authors should specify the dimensionalities of each variable and function in the paper. Unfortunately, none of them is clearly defined, which makes the technical part a bit hard to follow."
>
> We recognize that the first revision lacked precision in the introduction and Section 2. In the latest version, we now define all variables and functions (including the dimensions) explicitly. Please, see our general remarks and Section 2 in the manuscript.
>
>
> 2. "As a non-expert of the filed, I would like to know what exactly is a conformation x? Does it refer to the Euclidean coordinates of each node (atom) v of the graph?"
>
> In the introduction, we now define a conformation. We also replaced Fig. 1 with a new figure to help illustrate this difficult concept, especially for readers who are not familiar with chemical modeling.
>
> In the introduction, we now write:
> "Over the last few years, many highly-effective deep learning methods generating small molecules with desired properties (e.g., novel drugs) have emerged. These methods operate using graph representations of molecules in which nodes and edges represent atoms and bonds, respectively. A representation that is closer to the physical system is one in which a molecule is described by its geometry or conformation. A conformation $\mathbf{x}$ of a molecule is defined by a set of atoms $\{ (\epsilon_i, \mathbf{r}_i)\}_{i=1}^{N_v}$, where $N_v$ is the number of atoms in the molecule, $\epsilon_i \in \{\text{H}, \text{C}, \text{O},... \}$ is the chemical element of the atom $i$, and $\mathbf{r}_i \in \mathbb{R}^3$ is its position in Cartesian coordinates. Importantly, the relative positions of the atoms are restricted by the bonds in the molecule and the angles between them. Due to thermal fluctuations resulting in stretching of and rotations around bonds, there exist infinitely many conformations of a molecule. A molecule's graph representation and a set of its conformations are shown in Fig. 1."
>
>
> 3. "For each G, there are multiple possible conformations x_i, where each x_i specifies the Euclidean coordinates of all the nodes (atoms) v_j? Also, each node v_j has some attributes? So the goal is to represent each conformation x_i -implicitly- as a set of distances between the nodes v_j? I think that Sec 2.1. can be more clear about the definition of the data, such that to be accessible from non experts in molecule modeling. (Related to comment #1.)"
>
> We made significant changes to Section 2.1, including clear statements about the dimensionality of the node and edge features. Further, we believe that the new Fig. 3 will help the reader understand how the graph data flows through the model. Finally, you are right in thinking that nodes (which represent atoms) have attributes $v_i$ to encode the atomic element. Further, it is correct that it is our goal to model the distribution over distances between the atoms in the molecule.

---

> > ### Author Response · Authors · 2019-11-11
> > **Response to Review #2 - continued**
> >
> > 4. "I think that the definition of the model in Sec 2.2. could have been better. As before, I believe that it is hard to understand explicitly what the functions input and output is, since the dimensionalities are not defined. In my opinion, it would have been better if instead of this high level text you could provide explicitly all the steps of the model e.g. the concatenations, the appends, etc."
> >
> > We rewrote the section on the model definition (Section 2.2) to improve readability and precision. We now give details on the individual functions, their inputs, the dimensions of the output, and trainable parameters.  As stated above, we added Fig. 3, a figure that captures the details of our model and the general data flow. Further, we now provide all the hyperparameters of the model in the Appendix.
> >
> >
> > 5. "I could not understand if and how the model is able to handle graphs with different number of nodes, and different number of distances among them."
> >
> > The model can handle graphs with different numbers of nodes and edges because we are employing message passing graph neural networks (MPNN). In an MPNN, neighboring nodes exchange so-called messages over the edges connecting them to update their attributes. This network architecture is (in theory) applicable to any graph size or shape and has been applied to molecular graphs many times in the recent literature.
> > See for example:
> > Duvenaud, D. K.; Maclaurin, D.; Iparraguirre, J.; Bombarell, R.; Hirzel, T.; Aspuru-Guzik, A.; Adams, R. P. "Convolutional Networks on Graphs for Learning Molecular Fingerprints". In Advances in Neural Information Processing Systems 28; 2015; pp 2224–2232.
> >
> > In addition to the many changes we made to Section 2, we now also give a short overview of what an MPNN is:
> > "In short, an MPNN is a convolutional neural network that allows end-to-end learning of prediction pipelines whose inputs are graphs of arbitrary size and shape. In a convolution, neighboring nodes exchange so-called messages between neighbors to update their attributes. Edges update their attributes with the features of the nodes they are connecting. The MPNN is a well-studied technique that achieves state-of-the-art performance in representation learning for molecules."
> >
> >
> > We have updated our paper according to your review. We would appreciate it if you could let us know if you are willing to accept the paper now, and if not, let us know what we need to address/clarify further.
> > Thank you very much.

---

### Official Review · AnonReviewer1 · 2019-10-30
**Official Blind Review #1**

**Rating:** 6

**Review:**

The paper proposes a generative model for generating molecule with three dimensional structure. Given a graph, it leverages a variational autoencoder to embed the distance between two atoms into latent vectors (encoder) and then generate distance between two atoms based on the latent vector. Even if the problem is well motivated, I have several concerns:
I. What is the variable "x"? Is it possible to define a bit formally? It appears that x=(G,d) i.e. the graph with d.
II. It would be nice to develop the generative model of (G,d) i.e. both G and d. Simply generating d looks a bit restrictive.
III. How do you model/learn \mathcal{O}(x)?

I would be curious to know the answer of these questions.

**Experience Assessment:**

I have published one or two papers in this area.

**Review Assessment: Checking Correctness Of Derivations And Theory:**

I did not assess the derivations or theory.

**Review Assessment: Checking Correctness Of Experiments:**

I assessed the sensibility of the experiments.

**Review Assessment: Thoroughness In Paper Reading:**

I read the paper at least twice and used my best judgement in assessing the paper.

---

> ### Author Response · Authors · 2019-11-11
> **Response to Review #1**
>
> We thank you for your review and comments. We have incorporated all your suggestions in the latest revision of the manuscript.
>
> General Response:
> ================
>
> We made the following major changes to the paper based on the comments from all reviewers:
>
> 1. We improved the overall readability of the paper, especially for readers who are not familiar with the field of molecular modeling.
>    - We reworked the introduction to clarify the concept of a conformation.
>    - We replaced Fig. 1 in the introduction with a new figure that clearly shows how a molecule's conformations are related to its graph representation.
>
> 2. In Section 2.1, we expanded on the explanation of the molecular graph and the distances between atoms in a conformation.
>   - We now define all node and edge features formally.
>   - We modified Fig. 2 to illustrate how the set of distances d are extracted from the conformation x.
>
> 3. In Section 2.2, we significantly improved on the clarity of the explanation of the generative model.
>   - With additional equations and implementation details, the reader is now guided through the model and the overall data flow step by step.
>   - We added a new figure (Fig. 3) that shows the different components of the variational autoencoder and how the graph data flows through the model.
>   - We added additional information on how message-passing neural networks take graphs of arbitrary size and shape as input.
>   - All hyperparameters are now reported in the Appendix.
>
> 4. We simplified Section 2.4 on the proposed importance sampling scheme to improve readability and clarity.
>
> 5. We now propose a new benchmark (Conf17) instead of a new dataset.
>
> 6. We ran additional experiments:
>   - We studied the effect of the latent space size and report our results.
>   - We added ran simulations to add information on the computational effort of all methods.
>
>
> Individual Response:
> ==================
>
> I. "What is the variable "x"? Is it possible to define a bit formally? It appears that x=(G,d) i.e. the graph with d."
>
> In the introduction, we now clearly define what a conformation x is:
> "Over the last few years, many highly-effective deep learning methods generating small molecules with desired properties (e.g., novel drugs) have emerged. These methods operate using graph representations of molecules in which nodes and edges represent atoms and bonds, respectively. A representation that is closer to the physical system is one in which a molecule is described by its geometry or conformation. A conformation $\mathbf{x}$ of a molecule is defined by a set of atoms $\{ (\epsilon_i, \mathbf{r}_i)\}_{i=1}^{N_v}$, where $N_v$ is the number of atoms in the molecule, $\epsilon_i \in \{\text{H}, \text{C}, \text{O},... \}$ is the chemical element of the atom $i$, and $\mathbf{r}_i \in \mathbb{R}^3$ is its position in Cartesian coordinates. Importantly, the relative positions of the atoms are restricted by the bonds in the molecule and the angles between them. Due to thermal fluctuations resulting in stretching of and rotations around bonds, there exist infinitely many conformations of a molecule. A molecule's graph representation and a set of its conformations are shown in Fig. 1."
>
> We also replaced Fig. 1 to illustrate this concept.  Further, it is indeed possible to convert a graph G with the distances d back to a conformation x, as you assumed in your comment. This is exactly what our generative model does.
>
>
> II. "It would be nice to develop the generative model of (G,d) i.e. both G and d. Simply generating d looks a bit restrictive."
>
> It would be interesting indeed to construct a predictive model for both the graph G and the set of distances between atoms d. However, our model addresses a different question: Given a molecule G, what is the distribution over conformations p(x | G)? This is a pressing question in drug design, for example. Once a candidate drug is identified, properties need to be predicted for this molecule. This requires drawing samples from p(x | G) which is notoriously difficult. Our method attempts to address this very issue by providing a fast method that generates conformational samples in a one-shot fashion. We updated our introduction to clarify this.
>
>
> III. "How do you model/learn \mathcal{O}(x)?"
>
> In this study, we do not learn the function $\mathcal{O}(x)$ but employ a computational method based on quantum mechanics (density functional theory). The same applies to the function E(x).
>
> In the latest revision of the paper, we now state this explicitly in Sections 2.4 and 5.3:
> "... and computing $\mathcal{O}(x_i) \in R$ with a quantum-chemical method (e.g., density functional theory)."
>
>
> We would appreciate it if you could let us know if you have any other concerns or topics you would like us to address/clarify further.

---

### Author Response · Authors · 2019-10-03
**Sentence in introduction on existing machine learning methods needs to be more precise**

In the introduction, a sentence on existing methods based on statistical learning is not fully precise. The statement that they are not targeting a distribution over configurations but only the most stable folded configuration is not true for all of the methods. We will correct that at a later stage.

---

### Author Response · Authors · 2019-11-11
**General Response to Reviews**

We would like to thank all the reviewers for their thorough and constructive feedback. Below, we list major changes we made to the paper based on the comments from all reviewers. In the responses to the reviewers, we address their individual comments.

1. We improved the overall readability of the paper, especially for readers who are not familiar with the field of molecular modeling.
  - We reworked the introduction to clarify the concept of a conformation.
  - We replaced Fig. 1 in the introduction with a new figure that clearly shows how a molecule's conformations are related to its graph representation.

2. In Section 2.1, we expanded on the explanation of the molecular graph and how we obtain the distances between atoms in a conformation.
  - We now define all node and edge features formally.
  - We modified Fig. 2 to illustrate how the set of distances $\mathbf{d}$ are extracted from the conformation $\mathbf{x}$.

3. In Section 2.2, we significantly improved the clarity of the explanation of the generative model.
  - With additional equations and implementation details, the reader is now guided through the model and the overall data flow step by step.
  - We added a new figure (Fig. 3) that shows the different components of the variational autoencoder and how the graph data flows through the model.
  - We added additional information on how message-passing neural networks take graphs of arbitrary size and shape as input.
  - All hyperparameters are now reported in the Appendix (the code is available online).

4. We simplified Section 2.4 to improve the readability and clarity of the proposed importance sampling scheme.

5. We now propose a new benchmark (Conf17) instead of a new dataset.

6. We ran additional experiments:
  - We studied the effect of the latent space size and report our results.
  - We ran simulations to provide information on the computational effort of all methods.

---

### Decision · Program_Chairs · 2019-12-19

**Decision:**

Reject

**Comment:**

The paper presents a solution to generating molecule with three dimensional structure by learning a low-dimensional manifold that preserves the geometry of local atomic neighborhoods based on Euclidean distance geometry.

The application is interesting and the proposed solution is reasonable. The authors did a good job at addressing most concerns raised in the reviews and updating the draft.

Two main concerns were left unresolved: one is the lack of novelty in the proposed model, and the other is that some arguments in the paper are not fully supported. The paper could benefit from one more round of revision before being ready for publication.